# The Immune System—A Double-Edged Sword for Adenovirus-Based Therapies

**DOI:** 10.3390/v16060973

**Published:** 2024-06-17

**Authors:** Rebecca Wallace, Carly M. Bliss, Alan L. Parker

**Affiliations:** 1Division of Cancer and Genetics, Cardiff University School of Medicine, Heath Park, Cardiff CF14 4XN, UK; wallacer1@cardiff.ac.uk (R.W.); blissc@cardiff.ac.uk (C.M.B.); 2Systems Immunity University Research Institute, Cardiff University School of Medicine, Heath Park, Cardiff CF14 4XN, UK

**Keywords:** adenovirus, vector, immunity, T cell, antibody, seroprevalence, evasion

## Abstract

Pathogenic adenovirus (Ad) infections are widespread but typically mild and transient, except in the immunocompromised. As vectors for gene therapy, vaccine, and oncology applications, Ad-based platforms offer advantages, including ease of genetic manipulation, scale of production, and well-established safety profiles, making them attractive tools for therapeutic development. However, the immune system often poses a significant challenge that must be overcome for adenovirus-based therapies to be truly efficacious. Both pre-existing anti-Ad immunity in the population as well as the rapid development of an immune response against engineered adenoviral vectors can have detrimental effects on the downstream impact of an adenovirus-based therapeutic. This review focuses on the different challenges posed, including pre-existing natural immunity and anti-vector immunity induced by a therapeutic, in the context of innate and adaptive immune responses. We summarise different approaches developed with the aim of tackling these problems, as well as their outcomes and potential future applications.

## 1. Introduction

Adenovirus (Ad)-based vectors are popular tools for multiple clinical applications, as they are relatively easy to genetically manipulate, can be produced to high titers, and efficiently transduce cells. There is a broad range of Ad-based applications, with Ads developed as vaccine vectors, gene therapy vectors, and oncolytic virotherapy (OV) treatments. One of the most studied Ads in this context is human Ad type 5 (HAdV-C5), a species C Ad. However, to date, 7 different species (A–G) and over 50 distinct serotypes have been identified, which differ in structure, cell entry pathways, and pathogenesis [1,2,3,4].

There are a multitude of genetic adaptations that can be made to Ads. Deletion of genes essential for replication is one of the most common since it converts Ads from mildly pathogenic viruses to vectors that can no longer replicate or cause disease. This is particularly useful for vaccine applications, where short-term viral persistence is sufficient for the desired vaccine immunogenicity. Conditionally replicating Ads are useful as OVs, as they are designed to replicate only in malignant cells and leave healthy tissues unharmed. Other modifications can be made to retarget viruses to specific cells or tissues. HAdV-C5 uses multiple mechanisms for cell entry, and native Ad receptors often do not exclusively target the desired tissue, if at all [3,5,6,7]. Overcoming these native interactions is complex, although several approaches for Ad retargeting have been successful through both de-targeting of natural receptors [8] and re-targeting of vectors [5,7,8]. These recent advances make HAdV-C5 and other Ad vectors an even more intriguing tool for the development of precision targeted virotherapies.

The immune system is an additional limitation to the effective targeting of Ad vectors in vivo, particularly in systemic applications such as intravenous OV injections or administration of gene therapies. Following systemic application, the viral platform will encounter both the innate and adaptive arms of the patient’s immune system, which have evolved to be highly effective in clearing pathogens through numerous mechanisms [9].

Consequently, the ideal Ad-based vector will need to evade the patient’s immune system to permit safe and efficient treatment. In this review, we investigate various pathways through which an immune response against Ad vectors can be mounted, as well as their potential impacts on future treatments. Furthermore, we explore how research has approached these hurdles and critically appraise the advantages and disadvantages of each.

## 2. Basic Ad Structure

Although they can differ in immune epitopes, all Ads share the same basic structure. The ~35 kb double-stranded DNA genome and core proteins are surrounded by the major capsid proteins. The trimeric hexon protein is arranged in triangular planes, which form the faces of the icosahedral capsid. The vertices are formed by the integrated penton base, which connects to the trimeric fibre proteins, which consist of a shaft and a knob domain. Minor capsid proteins are integrated into the capsid structure [10,11] (Figure 1).

Comparing the amino acid sequences, the hexon proteins of different Ad serotypes mainly vary in the hypervariable regions (HVRs), which are structurally located on the outside of the capsid. The fibre differs between serotypes in the knob/receptor-binding regions, as well as in fibre shaft length [10,11,12]. 

All capsid proteins are involved in natural Ad tropism. The Ad penton base contains an RGD motif which binds integrins αvβ3 and αvβ5 [13]. These have an important role in Ad internalisation [14]. Commonly identified Ad receptors, which are usually bound through the trimeric knob domain, include Coxsackie and Adenovirus Receptor (CAR), CD46, Desmoglein 2 (DSG2), Sialic Acid (SA) and Heparan Sulphate Proteoglycans (HSPGs) [15,16]. 

CAR is the primary receptor for HAdV-C5 and HAdV-C2, as well as most species of C Ads [8,17,18,19]. As the name suggests, most Ads have some degree of affinity for CAR [18], although, interestingly, not all Ads use it as their primary receptor. Fibre length, as well as the fibre knob domains, differ greatly between species and change CAR binding capabilities [20]. For example, HAdV-D30 does not bind CAR, despite having CAR binding residues in the knob domain [21]. HAdV-D26 only weakly binds CAR, which has been attributed to sterical hindrance, limiting access to its CAR binding domain [22]. 

In HAdV-F40 and HAdV-F41, which express fibres of 2 different lengths on the same virion, fibre length has also been connected to CAR binding capacity. While the longer fibre protein is able to bind CAR, the more abundant shorter fibre binds HSPGs instead [16,23]. 

Species D Ads are highly diverse within their own species in the knob domain [22]. While they may engage CAR with varying affinity [17,24], SA may provide an alternative receptor target for some species D Ads [25,26,27]. Species D Ads may also enter cells via CD46; however, recent evidence suggests this to be via a direct interaction with hexon rather than via the fibre knob protein [28,29].

Receptor usage for species B Ads is similarly diverse. Species B Ad fibre knobs commonly bind CD46, for example, that of HAdV-B35 [30,31,32,33,34], whilst HAdV-B3, HAdV-B7, and HAdV-B11 appear to be able to bind both CD46 and DSG2 [35,36,37]. 

Similarly to the fibre, the hexon protein exhibits significant inter-species variation in the amino acid sequence. Most of this variation is restricted to the 7 HVRs located on the outside of the Ad capsid (Figure 2). Intra-species variability is also particularly high in species D Ads [38].

While the HAdV-C5 hexon protein binds blood clotting factor X (FX) in HVRs 5 and 7 [39], other species of Ads do not always have this interaction as they lack the critical binding residues. Using the amino acid identified as most critical to FX binding, we can predict this interaction across serotypes and species. From this, we can see that, while it is a common feature of Ads, species A and D viruses are not likely to bind FX (Figure 2). Similarly, species C Ads have also been shown to bind lactoferrin via HVR1 to enter CAR-negative cells via an as yet undetermined mechanism [40]. It has also been demonstrated that ChAdOx1 (which is derived from ChAd-Y25 isolate [41,42]), HAdV-C5, and HAdV-D26 may also weakly bind platelet factor 4 (PF4) via the hexon [43] in a charge-dependent manner.

Virus/host interactions dictate Ad tropism and toxicity. CAR is expressed at tight junctions between epithelial cells throughout the body [44], as well as on erythrocytes [45]. CD46 and SA are ubiquitously expressed. [46]. DSG2 is a transmembrane glycoprotein localised to desmosomes, junctions between epithelial cells of the heart, lymph nodes, and spleen [46]. FX binding of selected Ads will lead to liver transduction, as FX bridges the hexon-to-heparan sulphate proteoglycans (HSPGs) on the surface of hepatocytes [39,47]. There may also be more, yet unaccounted-for receptor/host protein interactions, which are related to specific Ad infective patterns and pathogenesis but have yet to be identified and explored. 

In addition to these interactions, which lead to cell entry and eventually infection, the virus will also be sensed by both the innate and adaptive immune system, eventually leading to the development of immunological memory. 

## 3. Basic Concepts of Antiviral Immune Responses

As an evolutionary necessity, the human body possesses an innate immune system, which acts as a first responder to any foreign attacks, such as injury or microbial infection. An immediate response initiated by the innate immune system is targeted at a broad range of pathogens. This response can be made more specific through the expression of pathogen-induced signalling molecules, such as type I interferons (IFNs). These are secreted by virus-infected cells upon the engagement of virus-specific pathogen recognition receptors (PRRs), which sense pathogen-associated molecular patterns (PAMPs). There are a multitude of cells involved in the innate immune response, which are activated through the secretion of type I IFNs [48,49] (Figure 3).

In the blood, non-cellular components of the innate immune system, such as complement, engage with pathogens to aid cellular components in mounting a response (Figure 3). Complement (C) proteins are found in the blood and activate the innate immune system via three pathways: classical, alternative, and mannan-binding lectin (which is specific to bacterial pathogens). The classical complement cascade requires antibody binding to a target to engage C1q. The alternative pathway does not require antibody binding but involves binding of C3 to conserved motifs of the pathogen directly. Once activated, complement has various mechanisms of pathogen destruction, such as facilitating destruction or direct lysis of cells through the membrane attack complex or facilitating opsonisation and subsequent clearance by phagocytic cells [51,52] (Figure 3).

Viral infection of cells also induces the expression of defensins, which are small peptides with anti-viral properties (Figure 3) [53]. Furthermore, upon the death of a virally infected cell, the engagement of PRRs with damage-associated molecular patterns (DAMPs) also contributes to the overall systemic state of inflammation [54].

Compared to the innate response, the response of the adaptive immune system is more specific to the pathogen encountered (Figure 4).

Broadly, there are two categories of cells involved in the adaptive response, though they can be further classified by receptor expression and specific function. These are categorised as responses driven by T cells, which mature in the thymus, and B cells, which mature in the bone marrow. Through maturation, T cell receptors (TCRs) specific to non-self antigens are selected and released to activate the adaptive response. T cell activation requires the presentation of antigens from the encountered pathogen through antigen-presenting cells (APCs) to naïve T cells [57] (Figure 4). Both T cells and B cells are able to differentiate into memory cells after a successfully mounted immune response, which allows for rapid expansion of relevant cell populations upon re-encounter of a pathogen [58]. 

B cell receptors (BCRs), much like TCRs, are matured in a rigorous selection process. Once formed, naïve B cells leave the bone marrow and await activation in the spleen. From there, they travel to secondary lymphoid organs, such as lymph nodes, where they sample passing antigens and await activation, which in viral infection is T cell-dependent. Activated CD4+ T cells recognizing a presented antigen on the B cells MHC-II will activate the B cell, causing it to rapidly undergo clonal expansion into plasma cells. These are able to secrete the former BCR as antibodies, a main effector of the adaptive immune response. Furthermore, they are able to specialise these antibodies through CD4+ T cells, helping to increase their target binding affinity [59,62] (Figure 4).

One of the most well-studied aspects of population anti-Ad immunity is neutralising antibodies (NAbs) due to their ability to directly inhibit Ad reinfection. Additionally, there are several other mechanisms through which antibodies can confer immunity to infection. There are 5 different classes of antibodies: IgM, IgE, IgD, IgA, and IgG, which are determined by the heavy chain structural part of the antibody. Plasma cells can be induced to switch antibody classes as needed by helper T cells; however, the epitopes recognised will not be affected by class changes. Different classes of antibodies are associated with different functions and different expression levels across tissues [60,62] (Figure 4).

Antibody effector functions include agglutination and opsonisation (crosslinking viral particles into larger aggregates and marking particles for uptake by phagocytes), as well as complement activation and neutralisation. While some of these require the presence of effector cells, many of these mechanisms can be facilitated by the presence of antibodies alone. Without effector cells present, the most common measure of antibody effect on viral pathogens is through neutralisation. This is not a singular process but sums up all mechanisms that can be involved in the reduction of infectivity [64,65].

The basic principles of antiviral immunology have been collected and extensively reviewed in more depth in the past [48,49,50,51,52,53,54,55,56,57,58,59,60,61,62,64,65].

## 4. Natural Infection

Ad infections usually cause mild and transient disease in healthy individuals. In rare cases, they cause more significant outbreaks (epidemics) in populations. In the mid to late 20th century, acute respiratory disease (ARD) caused by different Ad infections (HAdV-E4 and HAdV-B7) was common in US military recruits, which was successfully curbed by an emergency vaccination program using enteric-coated, live HAdV-E4 and HAdV-B7 as oral vaccines [66]. Since the discontinuation of the vaccine program in the late 1990s, cases of HAdV-E4 and HAdV-B7 reemerged in populations of military recruits [67]. Consequently, vaccinations were reintroduced in 2011 [68], and new vaccines are currently in development [69]. Severe Ad infections outside of these conditions have been rare and typically very restricted in the immunocompetent population [70]. In the immunocompromised, however, even a usually mild virus can cause significant harm [71]. Consequently, most Ads are originally isolated in hospitalised, immunocompromised individuals. Symptoms depend on the specific serotype, though some symptoms are common among Ads of the same species [1]. HAdV-D53, which was first isolated in 2005, causes keratoconjunctivitis [72]. Keratoconjunctivitis was also observed in HAdV-D37, isolated in 1976, which has also been found to cause rare cases of cervicitis [73]. In 2016 and 2019, HAdV-D8 caused several epidemic outbreaks of keratoconjunctivitis [74,75].

HAdV-D32 has been found to cause hepatitis, gastrointestinal (GI) ulcers, and respiratory disease [76]. HAdV-D15, first isolated in 1957, causes pharyngitis and respiratory and GI symptoms in different paediatric patients [77]. 

HAdV-B3 caused a severe outbreak of respiratory illness in a paediatric care facility in the US in 2005, with a 6% mortality rate [78]. An outbreak of HAdV-B7 was reported with 25% mortality at a paediatric care facility [79], and another outbreak of severe respiratory illness in a rehabilitation facility in 2018 was also attributed to HAdV-B7 [80]. 

More examples are summarised by Mennechet et al. [1].

### 4.1. Innate Immunity

Innate responses to acute virus infection are hard to measure and do not often appear in the literature. Mouse models often do not represent natural infection well, as the routes of administration are usually systemic, such as intravenous or intraperitoneal, which will result in less tissue-specific and more systemic responses. There are very few studies on acutely infected patients, which are needed to measure innate immune responses.

As described previously, Ad DNA can be sensed intracellularly once the virus has infected cells, leading to the release of IFNs. This is the main pathway for antiviral response in the innate immune system [81]. Doronin et al. suggest that induction of IFN signalling is dependent on the Ad’s ability to bind FX, as FX binding is essential for macrophage Ad sensing in the spleen. In this case, Ad-binding FX may serve as a PAMP, recognition of which is TLR4 dependent [82]. However, this study was conducted by intravenous administration of HAdV-C5 to mice and, therefore, may not be representative of other tissue-specific reactions. 

Macrophages are tissue-resident phagocytes and are very likely to encounter Ad particles upon infection by whichever route. In a macrophage cell line, cyclic GMP-AMT synthase (cGAS) was identified as one of the main drivers initiating the anti-Ad response. cGAS senses intracellular viral DNA and induces a signalling cascade resulting in the production of IFNs. Knockdown of cGAS, or any signalling molecule in the cascade, such as STING or IRF3, resulted in the loss of this function [83]. Plasmacytoid dendritic cells (pDCs) were shown to have a special role in the sensing of species B and other CD46 binding Ads through TLR9 sensing. Iacobelli-Martinez et al. showed that pDCs, which are found in secondary lymphoid organs such as lymph nodes, produced large amounts of IFN-α, while myeloid DCs sense Ads through intracellular DNA in a TLR-independent manner in donor peripheral blood mononuclear cells (PBMCs) in vitro [84].

Ads of different species may activate the innate immune system more or less strongly. Teigler et al. found that HAdV-B35 and HAdV-D26 induce much higher levels of pro-inflammatory cytokines, such as IFNs, in PBMC assays. They linked this to differences in endosomal escape. Compared to HAdV-C5, HAdV-B35 and HAdV-D26 release from the endosome much later, allowing for differences in intracellular receptor sensing, such as intra-endosomal engagement of TLR9 [85]. Johnson et al. also observed a much-increased production of IFN-α by HAdV-D28 and HAdV-B35 in their PBMC assays, compared to HAdV-C5, leading to a stronger activation of the NK response [86].

In a respiratory infection, as commonly found in species B Ads, resident cells in the lungs, respiratory tract, and local draining lymph nodes will be more involved in the infection. Fluid from bronchoalveolar lavage (BAL) samples in humans has been shown to contain defensins, which drastically reduced Ad infectivity in vitro [87]. Defensins confer neutralisation to Ad infections through intracellular degradation by preventing the fibre and pVI from detaching from the capsid, and, thereby, the virus escapes from the endosome [53,88]. Species D and F Ads show resistance to this pathway, which Smith et al. attributed to specific binding residues required in the penton base region, which are common to some but not all Ad species [53]. In serum samples of immunocompetent children in Argentina with acute respiratory HAdV-B infection of unspecified serotype, systemic levels of IL-6 and TNFα levels were associated with more severe disease. The study included 28 children, who were classified according to moderate, severe, and deadly infections [89]. Both IL-6 and TNFα are pro-inflammatory cytokines secreted by tissue-resident macrophages [90]. From this, we can assume that in most people, when Ad infection is transient and mild, the initial immune responses are mainly restricted to the infected tissue.

As Ad infections from various species cause GI symptoms [1], innate immune cells resident at these mucosal sites will be first responders during natural infection. Numerous immune cell types are present in the GI tract, as the body is commonly exposed to pathogens via this route. 

Some Ad infections, such as species D Ads (as well as some species B and E) often present as ocular infections [91]. While the eye is immune-privileged [92], the conjunctiva (the mucosal skin surrounding the eye), where Ad infection usually occurs, is not [1,91].

### 4.2. Adaptive Immunity

Many studies into population immunity against Ads use the neutralising ability of donor sera as a marker of anti-Ad immunity. In a large cohort of donors, neutralisation assays are relatively cheap and easy to run [93], especially in comparison to assays measuring T cells. 

While seroprevalence rates vary by location, such as HAdV-C5 being more prevalent in Europe than in Asia, there are also trends that can be observed across all regions. In 2003, Vogels et al. compared a number of rare Ad serotypes for prevalence in 6 different regions across the globe. While Species A and C generally had high seroprevalence, species B was far less common. Species D Ads, which are a much larger and more diverse species, showed lower seroprevalence rates than species A and C, though not as low as species B [94]. Another study by Abbink et al., published in 2007, examined seroprevalence in Africa, comparing HAdV-C5 to rare serotypes from species B and D, and found similar results. In their study, species D Ads were found to be less common than species B Ads, though both were significantly more rare than HAdV-C5 seroprevalence, which was observed in all samples [2]. Comparatively, in 2015, Dakin et al. investigated the seroprevalence of rare serotype HAdV-D49, a species D Ad, compared to HAdV-C5 in a Scottish cohort. HAdV-C5 neutralisation was found in 31% of samples, whilst NAbs against HAdV-D49 were not detected. [95] Another study by Klann et al. in 2022 tested the donor serum of a cohort in Germany for pre-existing anti-Ad antibodies across different species. Surprisingly, 11 out of 39 viruses were found to have higher seroprevalence than HAdV-C5, while the remaining 28 viruses were less seroprevalent. In agreement with previous findings, higher neutralisation was found in species C Ads compared to species B and D; however, NAbs against HAdV-B3, HAdV-C2, HAdV-E4, HAdV-C1, and HAdV-G52 were more prevalent than NAbs to HAdV-C5, which was unexpected [96]. 

NAbs can be directed against all major capsid proteins [97,98,99,100]. In serum samples from the United States and Africa, in 2005, Sumida et al. showed the presence of both fibre and hexon antibodies, though NAb titers against hexon were much higher [101]. Contrarily, Yu et al. found in a Chinese cohort that natural infection induces a strong anti-fibre response, as over 90% of their NAb serum donors exhibited antibodies binding the fibre, specifically targeting the fibre knob. They, however, confirmed that anti-hexon antibodies, though only found in about 40% of their donors, induced a much stronger neutralising response [98]. Bradley et al. also found that the main epitope targeted by NAbs in a South African cohort was the hexon protein, though fibre NAbs were also detected [97]. They also narrowed down the binding of hexon-specific NAbs to the HVR regions of the hexon protein [102]. The hexon HVRs are structurally located on the outside of the capsid, making them an ideal target for antibody recognition.

Though these studies all look at neutralisation, mechanisms of neutralisation may vary between different antibodies. An experimentally produced anti-HAdV-C5 hexon antibody, 9C12, was unable to prevent virus entry but did show neutralising activity. As the antibody remained bound after Ad internalisation, the authors proposed interference with Ad uncoating or replication [103]. The same antibody was later confirmed to elicit its neutralisation via blockage of microtubule trafficking after endosomal escape [88]. In another study, anti-HAdV-C5 antibodies protected virions from changes in pH, which are necessary for endosomal escape [104]. Anti-HAdV-C5 fibre antibodies have been shown to act mainly through agglutination [105]. Others showed that IgM anti-HAdV-C5 antibodies increase uptake and destruction by tissue resident macrophages, such as Kupffer cells in the liver [106], as well as in the blood [107]. As Ads are usually passed on through respiratory or GI infection, the mucosal antibody IgA may play a part in natural anti-Ad immunity. IgA can confer neutralisation through agglutination [108]. 

T cell responses, though just as important as antibody responses in the anti-Ad immune response, are investigated less commonly. PBMC assays are cost- and labour-intensive to run. As a recall response, CD4+ T cells require time for reactivation; CD8+ T cells can be reactivated within minutes, but only once the virus has entered cells and antigens are expressed in a MHC-I [109].

Olive et al. identified several T cell epitopes in conserved regions of the hexon protein. Epitope H910-924, located on the C-terminus of the protein, was shown to be particularly effective for developing CD4+ T cell responses [110]. H910-924 reactive CD4+ T cells were found in PBMCs of 14 out of 18 healthy volunteers and also predicted to be cross-reactive across a range of human as well as non-human Ads, such as chimpanzee, porcine, and bovine Ads [111]. Onion et al. performed PBMC assays on samples from 10 healthy donors and found 100% to have anti-HAdV-C5 specific memory CD4+ T cells. After in vitro stimulation of naïve PBMCs with HAdV-C5 whole virus, hexon, or fibre, responding T cells could be found to whole virus and hexon, but not fibre. This further indicates the importance of the CD4+ T cells in the anti-Ad response and hexon as an immunodominant epitope for CD4+ T cells. Upon expansion of anti-HAdV-C5 specific CD4+ T cells, they could also see cross-reactivity to other Ad serotypes and species [112]. Concurrently, a study in a Dutch cohort isolating PBMCs from 10 healthy volunteers found similarly high levels of CD4+ response, as well as cross-reactivity between human species A, B, C, and D Ads [113].

In studies into CD8+ T cells using PBMCs from seven donors, memory cytotoxic T cells that were stimulated in vitro were also able to cross-react between HAdV-C5 and HAdV-B35. When deployed against cells expressing hexon, penton base, and fibre antigen, they were most effective in killing cells expressing the hexon antigen, though anti-fibre and penton base responses were observed for some donors. Epitopes for CD8+ T cells were mapped to both the N- and C-terminus of the hexon protein [114]. Cross-reactivity of cytotoxic T cells was further established in other PBMC assays, where CD8+ T cells were isolated and successfully able to kill fibroblasts infected with HAdV-B7, HAdV-B11, HAdV-C2, HAdV-C5, HAdV-C6, HAdV-D8, and HAdV-E4, which once more was attributed to conserved hexon epitopes [115]. A more recent and much larger study involving PBMCs from 64 healthy donors and 26 immunocompromised patients with ongoing Ad infection revealed that the penton base, additionally to the hexon, contains immunodominant epitopes for CD8+ T cells. Using selected peptide sequences from both the hexon and penton base, strong responses could be elicited in donor PBMCs, with chosen hexon epitopes and whole virus stimulation eliciting stronger responses in CD4+ T cells and penton base epitopes alone leading to stronger CD8+ activation. Reactive cells to both hexon and penton base could be found across all donors, though a higher proportion of anti-penton base CD8+ T cells was found in patients with ongoing Ad infection [116]. The authors, therefore, conclude that higher numbers of anti-penton base T cells are required for control of an ongoing infection. However, as their patients are immunocompromised (and infection may not be well controlled at all), there could be other reasons involved, such as differing immune responses in the immunocompetent or less penton base-specific T cells being retained as memory cells. 

As described previously, both CD8+ and CD4+ T cells appear to primarily recognise more conserved regions of the Ads hexon protein [110,112,113,114,115] rather than the fibre and penton base protein [114,117], making them potentially cross-reactive between serotypes and species. This is true for memory cells as well [111]. While the response may be skewed towards CD4+ T cells [110,113], both play a critical role in the response to Ad infection. Hutnick et al. used PBMCs from 17 healthy donors to investigate pre-existing immunity against HAdV-C5, ChAd6, and ChAd7. Pre-existing CD4+ and CD8+ T cells to HAdV-C5 were detected in all donors and were also able to cross-react with ChAd6 and ChAd7, with which previous infection is extremely unlikely. Interestingly, while most CD4+ T cells displayed a memory phenotype, most CD8+ T cells displayed an effector phenotype instead. When tested for reactivity against specific peptides, CD4+ T cells were targeted to HVRs as well as conserved regions of the hexon. CD8+ T cells also responded against the hexon, however reactions were of higher magnitude compared to CD4+ T cells. The authors theorise that the high level of reactivity results from common contact with different serotypes of Ad, skewing selection for T cells against conserved epitopes, as well as low-level persistence of Ad in lymphatic tissue [118].

However, T cell responses can still vary between different donors and according to serotype. A 2005 study inducing anti-Ad memory in PBMCs found that the reactivation response against HAdV-C5 was very high across all donors, while responses against HAdV-A12 and HAdV-B11 were very low [117]. Unfortunately, PBMC assays may not always accurately represent in vivo conditions; consequently, more studies are needed to give a more in-depth understanding of T cell responses to natural Ad infection. 

As re-encounter of relevant antigens leads to an extension of protection by memory cells, continuous low-level or latent infection of wild type Ads below the threshold of clinical detection [119,120,121,122] may be an explanation for why anti-Ad memory cells can be commonly isolated from donor PBMCs [111,113,114,115,123]. Continuous low-level Ad infection has been proposed by Tran et al. to be related to the development of regulatory T cells (T-regs), which have immunosuppressive functions, allowing for low levels of Ad to evade the immune system [124]. Others have attributed the continuous low-level expression of anti-Ad antigens to the innate immune system instead, through successful suppression of Ad replication by type I IFNs [125]. While the suppression of an anti-Ad response vs. the development of an effective memory response may seem contradictory at first glance, the immune system is usually in a constant state of balance between pro- and anti-inflammatory responses. If low levels of Ad are able to evade immune evasion through the development of T-regs, the same low levels of antigen may still keep immunological memory sharp. 

In 2011, Adams et al. suggested that the CD46 binding capacity of HAdV-B35, while increasing DC engagement, simultaneously protected the virus from CD4+ T cell response. Direct binding of HAdV-B35 to CD4+ T cells via CD46 reduces the expression of pro-inflammatory cytokines, as has been shown for multiple CD46 binding Ads [126,127]. In naïve T cells, this mechanism impeded activation. Additionally, in activated and memory CD4+ T cells, cytokine production was also reduced as a result of infection. They suggest that this gives HAdV-B35, as well as possibly other CD46 binding Ads, an evolutionary advantage [126]; however, CD46 internalisation due to binding increases the likelihood of complement-induced lysis of cells [128]. This effectively trades lessened activation of the adaptive immune system to enhanced susceptibility to the innate immune system. 

In conclusion, due to the high level of pre-existing anti-Ad immunity in the population, any prospective Ad-based therapy already faces major obstacles. The presence of NAbs is the most immediate concern for intravenous applications, while applications that require long-term circulation or expression of Ads will also have to face broadly cross-reactive T cell responses. However, knowledge in this area is severely limited by the gap in knowledge between in vitro and in vivo responses. While the investigation of memory cell targets gives a decent picture of active Ad infection, as mentioned previously, PBMC assays may not always be representative of an in vivo system. Furthermore, since most current infections occur in the immunocompromised, measured responses of active infection may be different from how they would be in the general population.

## 5. Vector/OV-Induced Immunity 

Alongside anti-Ad immunity generated via natural Ad infection, anti-Ad immunity is also generated by the current use of Ad vectors in different therapeutic applications, such as recent mass vaccinations against SARS-CoV-2 using adenoviral platforms. Due to the scale of production and high replicative ability, the use of these vectors is on the rise. 

### 5.1. Widespread Ad Vector Development, Clinical Evaluation, and Use

As a potential vector for cancer and gene therapies, Ads, especially HAdV-C5, have been well investigated. The immunogenic cell death of tumour cells induced by Ad-based OVs and the release of DAMPs attracts the immune system to the tumour. The ability of the OV to replicate will allow self-amplification within permissive transformed cells over time without further administration. Genetic changes can be introduced to prevent Ad replication in off-target tissues or prevent replication altogether. The deletion of large parts of the genome, which usually completely prevents replication, allows for the insertion of large transgenes that can serve different purposes [129,130].

A very common example of creating conditionally replicating Ads is deletions in the early phase gene region 1 (E1). Whilst deletion of the entire E1 region prevents replication entirely [131], the deletion or loss of function of its genes E1B or E1A separately may be utilised in the context of OV cancer treatment [129,132]. In healthy cells, the E1 gene product will repress host genes involved in preventing viral replication, such as p53 and pRB. Both of these are very commonly mutated in cancers, allowing E1-deleted viral platforms to replicate conditionally in cells deficient in these pathways [129]. Additionally, replication, as well as transgene expression, can be placed under the control of a tumour-specific promoter [133], whilst structural changes to capsid proteins can be incorporated to retarget Ads from their natural tropism [8,133], making the vector even more tumour selective. 

Transgenes often incorporated into oncolytic Ad vectors include immunomodulatory genes to increase the patient’s anti-tumour immune response or prodrug converting enzymes to reduce the systemic toxicity of traditional chemotherapy drugs [133]. There are a number of oncolytic Ads currently being investigated in clinical trials, most of which are based on HAdV-C5, some of which are summarised in Table 1. 

Where long-term gene augmentation is required, such as with gene therapy for monogenic disorders, the use of helper-dependent or “gutless” Ad vectors, which are devoid of all viral DNA (except for the inverted terminal repeats, ITRs) and require a helper virus for production, has been extensively explored [146]. However, low-level contamination of helper Ads presents a problem, leading to the development of better purification methods [147,148] and, more recently, helper-free gutless Ads (HF-GLAds) [149]. These HF-GLAds have shown promise in pre-clinical testing but are yet to reach human trials [150].

Over the past decade, gene therapy applications have moved significantly toward using adenovirus-associated virus (AAV) vectors instead of helper-dependent Ads. AAVs have a smaller transgene capacity but allow long-term gene expression. They also tend to be less immunogenic than Ad vectors, which is desirable for long-term gene correction [151,152], though HF-GLAds and replication-deficient vectors also generate reduced anti-vector immune response compared to oncolytic or replication-competent Ads [150]. Some gene therapy applications of Ad vectors in clinical trials currently and in the past are listed in Table 2.

As vaccine vectors, Ads are generally used in a replication-deficient state. The historic vaccines against Ad, which worked extremely well, were live virus vaccines to protect military recruits against ARD caused by AdV-E4 and AdV-B7 [66]. For vaccination for any other purpose than Ad immunity however, production of viral proteins is a disadvantage, as the aim is for the immune response to be directed against the antigen transgene.

An interesting middle ground has been proposed by Crosby and Barry in their development of “single-cycle” (SC) Ads. Their vaccine vector, based on HAdV-C6, has a deletion in protein IIIa, which is essential for DNA packaging and virus assembly. The resulting vector, once it is inside the cell, is able to replicate its genetic material and, therefore, its transgene, without producing a viable progenitor virus. Some virus proteins are still produced by infected cells [161] and engage the immune system; however, the transgene/vector protein ratio is still more favourable. In a later in vivo experiment with Syrian hamsters to investigate the vector for influenza vaccinations, the single cycle replication vector required a 33 times lower dosage compared to replication-deficient vectors to achieve the same transgene levels, which is a drastically lower amount of vector protein patients would be exposed to [162]. Furthermore, the added Ad antigen may act as an adjuvant to recruit the immune system.

The value of Ad-based vaccines was demonstrated during the COVID-19 pandemic. According to ourworldindata.org, an estimated 5.13 billion people worldwide were vaccinated against SARS-CoV-2. In the UK alone, 50.76 million primary doses were administered to >75% of the population. Within the European Union, almost 100% of the vaccinated population was covered by 4 vaccine providers: Pfizer/BioNTech (Comirnaty), Moderna (Spikevax), Oxford/AstraZeneca (Covishield/Vaxzevria), and Janssen (Jcovden) in order of most administered [163] (Figure 5).

Both Pfizer (Comirnaty) and Moderna (Spikevax) are mRNA vaccines. The Janssen (Jcovden) vaccine is a HAdV-D26 vector-based vaccine expressing the SARS-CoV-2 spike antigens [164]. Similarly, the Oxford/AstraZeneca vaccine (Covishield/Vaxzevria) is based on their ChAdOx1 vector derived from ChAd-Y25, also with SARS-CoV-2 spike antigen transgene [42,165]. This means that in the EU alone, over 85 million people have been immunised with Ad-based replication-deficient vaccine vectors [163]. 

Although the SARS-CoV-2 vaccines were developed for intramuscular administration, different routes of administration have since been investigated for SARS-CoV-2 [166,167], in addition to other pathogens, such as influenza [168] and HIV [169]. Current uses of and clinical trials into Ad-based vaccines have been summarised in Table 3. 

As broad usage of Ad-based vectors becomes increasingly common, we need a solid understanding of administered vectors with patient immune systems to tailor the vector to be most effective in different applications. 

### 5.2. Innate Immunity

Overall, due to the limited receptors available, we would expect sensing by the innate immune system to be very similar between natural Ad infection and Ad vector administration, though it is possible that there are tissue-specific differences due to the differently localised or more systemic exposure to vectors, compared to natural infection.

In wild type Ads, the E3 protein enhances Ad immune evasion through several mechanisms, such as the prevention of intrinsic and extrinsic induction of apoptosis of virus-infected cells and the downregulation of MHC-I molecules on the cell surface [181]. While this leads to decreased engagement of cytotoxic T cells, it also means increased susceptibility to killing by NK cells due to the missing-self hypothesis [182]. Intranasal administration of HAdV-C2 and HAdV-C5 vectors with a complete E3 deletion to cotton rats led to an increased macrophage and polymorphonuclear leukocyte response. The same could be observed by only deleting the E3-19kDa region of the gene, making this the most important region in this response [183]. The E3 region is commonly deleted in many Ad vector applications, either fully or partially, for various reasons. Large deletions of the whole E3 gene region increase transgene capacity, while deletions of the E3-19kDa gene specifically can be used to actively increase antigen exposure, both of viral and tumour antigens in oncolytic vectors [129]. Examples of whole deletions include the oncolytic HAdV-C5-based Onyx-015 [184] or the very similar Oncorine, which is approved for oncolytic virotherapy in China [185]. Partial deletions have been made in several pre-clinical Ad vectors [8,186]. Though immune modulation may not always be the reason for an E3 deletion in vector development, it is important to take into consideration. 

Fejer et al. showed that, in intraperitoneal injections in mice, virus entry and endosomal escape trigger IFN expression in vivo in myeloid DCs in a TLR-independent manner and heavily reliant on the positive feedback loop of IFN secretion described earlier (Figure 3). They highlight the importance of splenic myeloid DCs in this interaction, as this is where the vast majority of IFNs were produced [187]. Vectors used in this study were based on HAdV-B3, HAdV-C2, and HAdV-C5 with no modifications to structural proteins. It is likely that the interaction of the Ad penton base with the integrin of the spleen is responsible for this strong response. Secretion of type I IFNs has been shown to induce the production of other anti-viral proteins in vitro, which inhibit viral DNA-, RNA, and protein synthesis [188]. In a vector context, this means reduced expression of transgenes through the induction of type I IFNs by the innate immune system. Di Paolo et al. also investigated the accumulation of their HAdV-C5-based vector in the spleen after intravenous injection of mice. They found that uptake by tissue-resident macrophages through integrin binding triggers the transcription of interleukin (IL) 1α within 10 min of administration, with high levels of expression induced 25 min after administration through positive feedback signalling. They also investigated the importance of integrin binding to the response and found a marked reduction in cytokine production if αvβ3 integrin cannot be engaged through the vector [189]. In a later paper, they outlined how the expression of IL1α leads to the expression of more chemokines, such as CXCL2, as well as complement proteins, which attract polymorphonuclear leukocytes to the area. They also observed the death of splenic macrophages, which, unlike the self-induced death of Kupffer cells in the liver, they were able to attribute to recruited targeted killing by recruited neutrophils instead [190]. This shows the importance of integrin binding in the very early innate responses to intravenously administered Ad vectors, as well as a potential mutation to bypass this mechanism through the removal of the integrin-binding RGD motif.

Kupffer cells are tissue-resident liver macrophages. As natural Ad tropism will direct intravenously administered Ad vectors to the liver, they are an essential factor in the innate anti-Ad vector response. While tissue tropism remains, Kupffer cell depletion markedly increases Ad vector gene delivery in intravenous delivered HAdV-C5-based vectors in mice [191]. Similarly, in vectors where Kupffer cell recognition is reduced, FX-dependent transduction of hepatocytes is markedly increased, as shown by Khare et al. in both intravenously injected mouse and Syrian hamster models [192]. Only the prevention of both FX binding and Kupffer cell uptake leads to an increase in systemic Ad delivery in vivo [191]. Interestingly, in repeat administrations within 72 h without prior Kupffer cell depletion, transduction to the liver and Kupffer cells is still reduced; instead, higher Ad gene expression is observed in the marginal zone of the spleen. Changes in biodistribution were attributed to Kupffer cell depletion after the initial dose due to immunogenic cell death. While the authors suggest Kupffer cells will have replenished by 72 h, these newly bone marrow derived cells will have slightly different receptor expression. Their impaired ability to clear Ad vectors leads to increased transduction to the spleen. At later time points of administration, Kupffer cells regain their scavenger receptors, and biodistribution is more similar to the initial administration [193]. This may be an important observation that may change considerations when designing repeat administration schedules, such as for oncolytic virotherapies. 

Interactions with components of the complement system further activate the anti-vector immune responses. Ad vectors can directly interact with C3 [194]. In the absence of C3, intravenously administered HAdV-C5 vectors cannot induce thrombocytopenia in mice, indicating that complement is involved in this response. Overall cytokine release and systemic inflammation were also reduced in C3 knockout mice [195], which can be expected as C3 is involved in positive feedback loops activating the innate immune system (Figure 3). Tian et al. observed that in vivo in mice after intravenous application, complement mainly senses Ad vectors when they have already infected cells, as an HAdV-C5 vector defective in cell entry had a much-reduced complement response. In vitro, complement was able to interact with vector particles directly, causing a much larger response [196]. This is important to note, as it demonstrates that data on complement will likely have to be collected using in vivo experiments to provide accurate results. For example, the Ad hexon has also been implicated in binding IgG, which may further facilitate complement binding and neutralisation via the complement cascade. In vitro, this mechanism reduces Ad transduction via destruction by the classical complement cascade. In vivo, there appear to be other mechanisms involved as well [196]. As described earlier, signalling of activated cells of the innate immune system can increase complement activity (Figure 3), which would not be picked up by pure in vitro experiments.

As with natural Ad infection, vectors from different serotypes and species will engage the immune system in different ways, which may make some serotypes more suitable than others. As touched upon before, different species activate the innate immune system to different levels due to differences in tricellular trafficking in natural infection. This is also true for Ad vectors. In a rhesus monkey vaccination study by Teigler et al., low seroprevalence vectors based on HAdV-B35, HAdV-D26, and HAdV-D48 induced a much stronger innate cytokine response than the HAdV-C5 vector [197]. 

Though the innate immune system presents challenges for the immediate systemic administration of Ad-based vectors, the innate and adaptive immune responses are intrinsically linked. Differences between vectors in the innate response will likely also translate to differences in the adaptive response.

### 5.3. Adaptive Immunity

The development of anti-vector NAbs is a concern for Ad use in any vector context. Neutralisation occurs instantly when viral vectors are administered. There are several mechanisms through which anti-vector antibodies can neutralise an administered Ad-based therapy. Anti-vector antibodies can redirect Ads to phagocytes, such as macrophages, and inhibit endosomal escape, leading to Ad degradation rather than propagation in mice [107]. Aggregation has been suggested as the main mechanism of anti-hexon neutralisation in intramuscularly vaccinated rabbits, though the same article also confirms post-entry neutralisation of Ads [104]. This is very likely through a similar mechanism as described above, involving intracellular actions of antibodies. For vector-specific anti-Ad antibodies, the intracellular receptor TRIM21, which is antibody activated, has been shown to be a major facilitator of neutralisation through the recruitment of proteases to digest intracellular vector particles [198].

Using the same HAdV-C5 vaccine vector encoding different transgenes does not give a satisfactory anti-transgene response, as demonstrated by Steffensen et al., which they were able to attribute mainly to the development of anti-vector NAbs. However, priming and boosting against the same antigen using the same vector is possible, likely because a much lower concentration of antigen is required to activate the recall response [199]. 

Intratumoural injection of replication-deficient Ad vectors into subcutaneous tumours in mice generates a strong systemic anti-vector NAb response, which is an important consideration in potential Ad-based cancer therapies [200]. In 1998, Gahéry-Ségard et al. identified in intratumoural injection of replication-deficient HAdV-C5-based vectors in 4 lung cancer patients that antibodies against fibre are expressed first, followed by anti-penton base and anti-hexon antibodies. While half the patients developed a strong anti-penton base IgG response, the other half developed a strong anti-hexon IgG response. They also found that the presence of anti-penton base antibodies was indicative of a strongly neutralising anti-vector response [99]. More recent studies into oncolytic vectors based on HAdV-C5 found that, while replication-deficient vectors were suppressed more by NAbs (in both human and mouse serum), oncolytic vectors still had diminished transduction in cancer cell lines. Cell killing in vitro was also markedly reduced by the presence of high titer NAbs. In the presence of low titer NAbs, cell killing was possible through overcoming NAbs, though it was inhibited at earlier time points. Consequently, they found that the higher the NAb titer, the later (if at all) decrease in cell viability was detected. In subsequent in vivo studies with tumour xenografts in mice, transgene expression was reduced after the administration of anti-vector NAbs to nude mice. Tumour shrinkage was observed in mice with a low dose of transferred anti-Ad NAbs but significantly reduced in mice transferred with a high dose of NAbs [201]. Since for xenograft experiments, mice need to be immunosuppressed, the authors transplanted high or low amounts of NAb serum from other mice induced by intravenous challenge, then injected Ad therapies intratumorally. This excludes the presence of cellular immunity, so any effect will be due to NAbs only. While informative, that means these results are likely not representative of a full in vivo system. 

Intravenously administered replication-deficient vectors induce both anti-hexon and anti-fibre antibodies in mice, though anti-hexon antibodies have a much stronger neutralising effect [97]. The main target for this neutralising response is the HVRs of the hexon [102]. 

A vaccination study in healthy volunteers by Cheng et al. showed that intramuscular vaccination induced NAbs against capsid proteins other than the fibre [202]. Previous in vitro reports indicate that this may be due to an anti-hexon response [104,203]. In intramuscular vaccinations with chimeric vectors based on HAdV-C5 and HAdV-B35, which previously showed to be non-crossreactive, NAbs against both parent vectors were detected. However, vector-specific anti-hexon NAbs were more common and effective in adoptive transfer studies using purified antibodies from previously vaccinated mouse serum [101]. Following a Phase II intramuscular vaccination trial using a HAdV-C5-based vector, NAbs to vector pre- and post-vaccinations were measured, only distinguishing between fibre and not-fibre. Vaccine-induced NAbs were directed against the capsid and, in some cases, both capsid and fibre, while fibre NAbs were mostly detected in volunteers that were seropositive for HAdV-C5 pre-vaccination [204].

Steffensen et al. compared subcutaneous and intramuscular routes of administration using the same replication-deficient vector expressing different transgenes and found that the route of administration generating the neutralising response did not matter [199]. More recently, the ChAd36-based vaccine vector BB154 (Table 3) showed that after intranasal vaccination in Phase III clinical trials, vector-specific neutralising antibody levels were low [167]. On the other hand, a Phase I clinical trial utilising HAdV-C6 and ChAd3 vectors as intramuscular Hepatitis C vaccines in the UK found heterologous prime-boost schedules to be less effective than expected. They attributed this to cross-reactive NAbs, effective against both vectors [205]. This highlights how differences in the route of administration and different serotypes can alter the immune response to the vector, which, if harnessed correctly, can work to our advantage.

Another Phase I clinical trial using a ChAd63 vaccine vector intradermally and intramuscularly measured anti-ChAd63 NAbs pre- and post-vaccination. Pre-vaccination NAbs to the vector were low (17% of volunteers) but present. These are likely cross-reactive responses from previous exposure to human Ads, with HAdV-E4 being the closest related human Ad to ChAd63 [206] (Figure 1 and Figure 2). The study concluded that since participants with pre-existing anti-ChAd63 NAbs were not observed to mount a reduced response against the transgene, this will likely not prevent future applications [207]. Antibody titers pre-vaccination (i.e., natural NAb titers) were substantially lower than post-vaccination titers. Higher post-vaccination NAbs could impede the use of the vector; however, when volunteers were again vaccinated with ChAd63 against the same antigen over 6 months later, T cell responses did re-boost. Antigen appears to play a key role here, as re-vaccinating against the homologous antigen in a subsequent ChAd63 trial demonstrated re-boosting of T cell responses and maintenance of antibody responses against the transgene [208]. However, this does not prove that the vector can be re-used to vaccinate against heterologous antigens in the presence of heightened anti-vector immunity. 

Further, Ewer et al. and Kimani et al. reported on the same ChAd63-based vector in the UK and African cohorts of a Phase I/II trial. Both measured low levels of pre-existing NAbs to ChAd63 [209,210]. Interestingly, while Ewer et al. noted a trend toward higher T cell responses to the insert in those with anti-ChAd63 antibodies [209], Kimani et al. observed the opposite: higher anti-vector NAbs correlated with a slight reduction in T cell anti-transgene response [210]. 

Vaccinations with the ChAdOx1 vector described earlier have put the spotlight on the reusability of the vector. A Phase 2/3 COVID-19 vaccine trial intramuscularly administrated ChAdOx1. Anti-vector antibodies were induced with the first vaccination; however, boosting did not increase anti-vector NAbs further. The presence of anti-vector NAbs pre-boost vaccination had a minor negative impact on anti-transgene NAb titers but did not negatively impact the anti-transgene T cell response [211]. As mentioned previously, boosting against the same antigen is possible despite the presence of anti-vector immunity. It is likely that repeated boosters using the ChAdOx1 vector against the same SARS-CoV-2 antigen will be successful. However, comparing the same metrics (prime vs. boost NAbs against transgene or prime vs. boost cellular response against transgene) might have been more informative. 

In 68 ChAdOx1 nCoV-19 vaccinees in Korea, pre-existing anti-Ad NAbs were detected in 49 volunteers. This is not surprising, as pre-existing anti-Ad immunity is very commonly found in populations. The transgene response was observed to be much higher overall in vaccinees without pre-existing anti-Ad NAbs, especially after booster vaccine [212]. This is very surprising, given that it is unlikely that many people will have specifically encountered a ChAd before, but not unique as pre-vaccination anti-ChAd antibodies have been measured in other studies [207]. The authors of the Korean study state that this may be due to the presence of cross-reactive NAbs. 

Using a different vector based on HAdV-C5, 346 Mexican subjects were vaccinated intramuscularly with Ad5-nCoV. Hernandez-Bello et al. observed a trend that pre-existing anti-HAdV-C5 NAbs led to a less effective transgene response, but this result was not statistically significant [213]. This again defies expectation, as anti-HAdV-C5 NAbs are usually very common. HIV vaccine studies tested intramuscularly in rhesus macaques based on HAdV-C5, ChAd6, and ChAd7 showed no anti-Ad antibodies before the first dose. Low titer NAbs to ChAd7 were found after the first dose, with a slight increase over time, while titers of NAbs to HAdV-C5 were high even within a short time after first administration. Booster with a different Ad serotype not only produced vector-specific NAbs but also increased the amount of anti-prime vector NAbs [214]. This is a good example of how different types of Ads can be more or less immunogenic.

Overall, anti-vector NAbs pose a problem to both vaccine and OV applications, especially upon repeat administrations, as NAbs often show the strongest activity against vector-specific regions such as the hexon HVRs. However, there is also some evidence for NAbs cross-reacting even across species. It is possible that, by only looking at antibody responses, some studies attributed cross-reactivity to NAbs when cellular immunity may have played a part in unexpected responses as well, especially in human subjects with pre-existing anti-Ad immunity. 

Trials for a HAdV-C5-based SARS-CoV-2 vaccine in China results showed that vaccine immunogenicity is limited by pre-existing anti-HAdV-C5 immunity [215], though authors do not expand on whether this effect is largely due to T cell immunity or neutralising antibody responses. Likely, both play a role. It has been shown that pre-existing cross-reactivity of T cells between Ad serotypes can impair the generation of anti-transgene antibodies, even in rare ChAd-based vaccine vectors [208]. 

Anti-Ad vector T cell responses and epitopes are not often studied in vivo. Instead, most articles focus on differences in anti-transgene responses as a measure of efficacy for new vaccine vectors [216,217,218,219,220]. Whilst responses to transgene are important, the development of anti-Ad vector responses and how they differ according to route of administration needs to be studied in more detail.

Heemskerk et al. investigated anti-Ad T cells in PBMCs from 25 healthy donors. In response to stimulation with inactivated HAdV-C5, 76% of donor T cells reacted. Newly stimulated T cells were found to be cross-reactive to varying degrees: 3 clones were cross-reactive within species, 4 clones between some species, and 5 cross-reacted between all species tested [221]. While T cells can often be widely cross-reactive, this shows that vector-specific T cells are also raised in response to vector administration. Schirmbeck et al., similar to other studies, found no response against a new transgene in intramuscularly HAdV-C5 vector vaccinated mice induced with pre-existing Anti-HAdV-C5 immunity. Their experiment showed that viral replication was not required for the development of an anti-vector CD8+ T cell response, so minimal amounts of antigen are required for an effective response, which was determined to be targeted to highly conserved regions of the hexon [222].

Investigating the low seroprevalence ChAd63 as a vector for intramuscular malaria vaccines in a Phase I clinical trial, pre-existing T cell responses to the ChAd63 vector hexon were found in 8 out of 12 volunteers tested and increased in all participants after vaccination. Pre-vaccination anti-vector T cell responses led to lower anti-transgene antibody responses. Both CD4+ and CD8+ T cells were found to contribute to this anti-vector response, with CD4+ T cells likely underpinning the negative correlation [208]. As stated previously, it is unlikely that participants encountered ChAd63 through infection; hence, the observed response likely resulted from cross-reactive T cells mounted in response to other Ad infections. 

Another Phase I clinical trial in the UK investigated using HAdV-C6 and ChAd3 vectors as heterologous HepC vaccine vectors. The study excluded volunteers with anti-vector antibody titers of >200 to prevent NAbs from interfering with vector immunogenicity. They also measured anti-vector T cell response against the hexon. Responses were found even in non-primed subjects and were increased substantially after vector administration. Booster vaccinations induced less transgene response than predicted. The study excluded participants with >200 anti-Ad antibody titers. Authors concluded that anti-hexon T cell responses were actually positively correlated with transgene response and instead attributed less-than-expected booster responses to cross-reactive NAbs developed after the first administration [205]. Despite measuring a substantial increase in vector-targeted T-cells, authors attributed low booster responses to NAbs. It is extremely unlikely that T cells are completely uninvolved in the anti-vector response presented, although it is possible they could have a positive impact. 

A large amount of anti-Ad vector data is available from the STEP trial (NCT00125970) and follow-up data analyses. A replication-deficient HAdV-C5 vector was used in a phase II clinical trial as an HIV vaccine vector. A total of 480 participants were vaccinated intramuscularly. The trial failed to protect vaccinees from HIV infection and actually made some recipients of the HAdV-C5-based vector (uncircumcised men) more susceptible to HIV. While this was a devastating result, it led to several more in-depth studies on anti-Ad vector responses and how these may affect vector applications. Using PBMCs from trial participants, the HAdV-C5 vector was found to induce receptor expression favourable to HIV in CD4+ T cells, as well as inducing a stronger anti-vector CD4+ than CD8+ T cell response, making them more susceptible to HIV infection [223]. In a follow-up from the STEP trial, Frahm et al. investigated cellular immunity to Ad vectors. Ad-specific T cells could commonly be detected even in the absence of anti-Ad NAbs (in 54% of the seronegative/placebo-vaccinated group). Pre-existing CD4+ and CD8+ T cell immunity to Ads induced impaired anti-transgene responses in vaccine vectors, regardless of the presence of NAbs. T cell target epitopes map to the conserved region of the vector Ads. For CD4+ and CD8+, most of these are commonly hexon, but other, non-structural epitopes are also recognised, such as protein (p) V, pVII, E2 terminal protein (E2pTP) for CD4+ and E3 glycoprotein 19 (E3gp19) and E4 open reading frame 6 (E4ORF6) for CD8+. Interestingly, E3gp19 and E4ORF6 were only targeted in the vaccine group, though results were not statistically significant due to the small sample size [224]. Hutnick et al. identified pre-existing anti-Ad CD8+ T cells in 95% of unvaccinated volunteers of the STEP trial. They attribute these high numbers to cross-reactivity rather than the natural occurrence rate of HAdV-C5. Differences in CD8+ response were observed between different NAb titer subjects, though anti-vector response overall increased regardless of seropositivity before vaccination, even with replication-deficient Ads [225]. Overall, these results emphasise that serotype-specific immunity, though important more immediately upon administration, cannot predict later cross-reactivity due to T cells, resulting in potential loss of efficacy of vector applications. Furthermore, seropositivity, as defined by NAb titer, is not an indicator that a person is completely naïve to Ad infection. More research into anti-Ad vector responses is required to more accurately predict how a vector is likely to perform in the clinic. Furthermore, it is impossible to regard only one side of the anti-vector response, as T cell and B cell responses (as well as the innate immune system) are intrinsically linked.

For example, early gene therapy studies for the treatment of cystic fibrosis in mice using an intranasally administered HAdV-C5-based vector highlighted the importance of CD8+ T cell anti-vector responses, as mice that were challenged pre-treatment and only able to mount CD4+ T cell responses still saw some transgene expression. Additionally, the development of NAbs to the vector made subsequent administrations much less effective, even in the absence of T cells [226]. Sumida et al. studied intramuscular immunisations with HAdV-C5-based vectors. After immunisation, they transferred splenocytes and serum to naïve mice to separately test the effects of T cells and NAbs on subsequent vector administration. Both resulted in drastically lower responses to the transgene. Transfer of high levels of NAbs resulted in immediate but short-lived vector inhibition. A total of 53% of mice were still able to induce a transgene response after the transfer of splenocytes from double vaccinated mice, though anti-vector antibodies were developed after 2 weeks, either through transferred B or CD4+ T cells. This elegantly shows the synergistic effect of antibodies and T cells: while NAbs confer immediate immunity, long-term immunity requires CD4+ T cell help. The immediate vector suppression from the transfer of splenocytes was attributed to CD8+ T cells [227]. Similarly, the adaptive immune system will always interact with the innate immune system. For example, complement protein C3 is required to mount a neutralising anti-Ad vector antibody response, as found by Appledorn et al. [228]. The complement system is an important bridge between innate and adaptive immunity, and many of the cytokines secreted due to complement activation are essential for the maturation of T and B cells. In the absence of these pro-inflammatory cytokines, it is possible to generate a tolerating anti-Ad response instead of a neutralising one.

As briefly touched upon before, different species and serotypes of Ads are considered to have different immunogenicity [2,41,229]. Johnson et al. showed that in PBMCs from healthy donors, HAdV-D28 and HAdV-B35, lead to increased death of APCs. Consequently, less antigen was presented to T cells, and response to the transgene was significantly lower than equivalent HAdV-C5-based vaccine vectors [86]. While in a vaccine context, this is not ideal, it also means decreased presentation of vector antigens, leading to reduced anti-vector responses and increasing the potential for effective repeated administration. Similarly, a study investigating a broad range of ChAd vectors as well as a selection of rare HAdVs for their suitability as vaccine vectors in vivo, found lower doses of species C Ads were required to induce a robust anti-transgene response compared to other human Ads in mice and rhesus macaques [230]. This could be attributed to a similar mechanism as Johnson et al. In the same study, ChAd vectors were surprisingly effective at inducing anti-transgene responses and, additionally, were found to be minimally affected by cross-serotype and cross-species anti-vector immunity in mice [230], which indicates no cross-reactive neutralising antibodies are produced. Responses to the same ChAd vector subsequently carrying different transgenes, however, were very limited [230], indicating a strong vector-specific response was still induced. 

In some examples, Ad hexon PF4 binding has been linked to very rare instances of thrombotic thrombocytopenia [43,231,232], which was only apparent due to the wide rollout of the ChAdOx1 nCoV-19 (Covishield/Vaxzevria) and Ad26.COV2.S (Jcovden) vaccines during the coronavirus pandemic. Vaccine-induced thrombotic thrombocytopenia (VITT) is hypothesised to be driven by autoantibodies against PF4 [232]. It is extremely rare, with roughly 1 case occurring in 67,000 primary dose vaccinations and 1 in 500,000 booster dose vaccinations with ChAdOx1 nCoV-19 in the UK, according to the National Institute for Care Excellence (NICE) [233,234]. Interestingly, the incidence of VITT in the US after administration of the Ad26.COV2.S vaccine was estimated to be 1 in 263,000 administered doses, which is lower than was found with the ChAdOx1 vector [235]. The generation of autoantibodies to PF4 induces aggregation, leading to platelet activation and resulting in a thrombotic event [43,236,237]. The binding of PF4 to the viral vector, which may well be critical in the generation of anti-PF4 autoantibodies, has been suggested to occur via HVR1 [238] and involve interactions at the inter-hexon spaces [239], though neither of these theories have been confirmed. Confirming this mechanism would also allow for predictions of other Ad serotypes to bind PF4 and their consequent prevention. 

## 6. Engineering Ads for Immune Evasion

Mechanisms described previously are important in protection from pathogens. In the case of vectorised Ads, however, they might hinder treatment as the immune system is unable to distinguish between the pathogenic virus and the therapeutic vector. Ad vectors are in high demand for multiple clinical applications, and various strategies have been explored to reduce the effect of anti-Ad immunity, both natural and vector-induced, on potential future treatments.

In numerous Ad-based gene therapy platforms, a decline in gene expression over time is observed [240,241,242], and integration of the target gene needs to be incorporated to increase the duration of gene expression [146]. Natural hepatocyte turnover and expression of vector antigen were suggested as causes for long-term failure in a gene therapy study into haemophilia B in mice, which spanned one year [240]. In cystic fibrosis, which was a popular target for initial investigations into vector-based gene therapies due to the ease of access to the most affected tissues, the initial benefits of vector administration quickly deteriorated due to the strong anti-vector response [243]. This fact was likely aggravated by the fact that the study design called for multiple vector administrations, and the same HAdV-C2-based vector was used for all of them. Brunetti-Pierri et al. achieved transgene expression of a helper-dependent Ad vector for as long as 7 years in baboons, though gene expression still declined by about 10% per year [244]. In order to not lose efficacy over time, the development of neutralising and memory responses must be minimised.

In cancer therapies, where replicating vectors are used, the problem is not just presented by reaching the target site but also maintaining the levels of the oncolytic vector high enough to have a therapeutic effect. As oncolytic Ad vectors are able to replicate in cancer cells, this increases exposure of the vector to the patient’s immune system. In a single application, the engagement of the immune system can serve the purpose of immunogenic cancer therapy. The release of viral epitopes and PAMPs can serve as an additional activating factor for the anti-cancer immune response desired in oncolytic and can help flag infected cancer cells for destruction to both the innate and adaptive immune systems [245,246]. However, where additional, subsequent doses of oncolytic are required, this response may work against the patient instead, as the development of vector-specific neutralising antibodies will prevent effective systemic administration. Consequently, for oncolytic Ad vectors, a fine balance needs to be found between engaging the immune system in a beneficial way and evading anti-vector immunity where it does not serve the patient. 

For Ad-based vaccines, boosting against the same antigen appears effective, with response to the antigen improved further when using a different vector in a heterologous booster vaccine [247]. This may be due to anti-vector immunity impeding the booster dose. The beneficial effect of heterologous vs. homologous vaccine strategies is well established and could be attributed to anti-vector immunity [248,249]. The reduction in protection is shown to be even more pronounced when vaccine recipients were tested for anti-Ad antibodies prior to vaccination. In a trial for an Ad-based Ebola vaccine in 2010, seronegative participants had a significantly better antigen-specific antibody response than those that were seropositive for HAdV-C5 [250]. This has been seen in multiple other HAdV-C5-based vaccines since [251,252,253]. However, anti-vector immunity may not always be undesirable. The immune activation against the viral vector may reduce the need for adjuvants and boost the response to the target epitope [248]. 

To maximise the efficacy of Ad-based therapeutics, we require Ad vectors that are able to evade aspects of all of the presented pathways, as all of them will universally play a role in the response to administered vectors [227]. However, as described, there is no one-size-fits-all solution to the issue, as different vector applications may benefit from different solutions.

Here, we summarise these attempts in 3 categories: structural modifications, chemical shielding, and other approaches (Figure 6).

### 6.1. Structural Modifications

Ad vectors based on different species than HAdV-C5 have been shown to be beneficial in evading pre-existing immunity to HAdV-C5. Some groups have been successful in utilising non-human Ads, such as ChAds, bovine, or porcine Ads. As they are rare or even absent in the human population, there is less likelihood of specific immunity to whichever Ad is chosen [249,254]. Cross-reactivity between different serotypes, especially when originally targeting different hosts, is also usually low [254,255]. A 2021 study in rhesus macaques using a ChAd36-based vector found that a single intranasal vaccination dose conferred protection against intranasal and intrabronchial challenge with SARS-CoV2 [256].

Within the human Ads, vectorising alternative species to HAdV-C is a popular strategy [257,258], especially looking at species B [94,259] and D [258,260] Ads due to their particularly low seroprevalence [2,94,95,96].

Vectors that successfully evaded pre-existing anti-Ad immunity have been designed based on HAdV-D26 for the Jcovden SARS-CoV-2 vaccine [261], HAdV-B11 [259] and HAdV-B35 [94,262] in various applications. Enadenotucirev, or EnAd (previously called ColoAd1), took the approach of directed evolution to produce a novel tumour-selective Ad vector where replication is selective for tumour cells. The resulting vector contained parts of HAdV-B3 and HAdV-B11p and successfully evaded anti-HAdV-C5 antibodies present in human serum, neutralising HAdV-C5 [263]. However, the repeated administration of the same vector still carries the drawbacks of anti-vector immune responses, potentially making this approach self-limiting [254,255]. 

Penaloza-MacMaster et al. investigated T cell responses to vectors based on HAdV-D26, HAdV-B35, and HAdV-D48. The advantages of their chosen vectors in vaccine applications included less exhausted T cell phenotypes and quicker memory reactivation to the transgene [219]. Unfortunately, they did not look into T cell responses against their vector. If they mirrored the response to the transgene, this would likely heavily impair repeated vector administration. 

Investigations into HAdV-D28 and HAdV-B35 as alternative vectors to HAdV-C5 revealed that they induce an increase in NK cell response, reducing their potential for transgene expression in infected cells [86]. This may impair their uses as vaccine vectors, where the death of transfected cells is unwanted. However, it may make them well suited to applications such as oncolytics, as they are rare serotypes able to engage in killing through the innate immune system, but through killing of APCs may not induce as much adaptive anti-vector response.

A comprehensive study by Wang et al. in 2023 evaluated 39 different serotypes covering all the Ad human species for their potential suitability as vaccine vectors. They specifically focused on CD8+ T cell and DC activation, as well as cytokine expression in their in vitro assays, and predicted HAdV-C1, HAdV-D8, HAdV-B7, HAdV-F41, HAdV-D33, HAdV-C2, HAdV-A31, HAdV-B3, and HAdV-D65 to be particularly useful if vectorised [229]. As they were looking for high immunogenicity in potential vaccine vectors, it will be interesting to see if these predictions hold true and if viruses they scored with lower immunogenicity ratings may be advantageous in engineering vectors with increased immune evasion in the future.

Several studies have investigated implementing structural changes to the Ad major capsid proteins. Wholly or partially exchanging the hexon protein with the externally exposed HVRs from low seroprevalence has provided these recombinant, chimeric vectors with protection from neutralising anti-HAdV-C5 antibodies.

Whole hexon swaps, especially between Ads of different species, are often structurally incompatible, as shown by Youil et al. [264]. Of 18 attempted structural changes, only 4 produced viable vectors. Transduction efficiency in vitro was reduced in HAdV-C5/C1 and HAdV-C5/C2 and drastically reduced in the HAdV-C5/A12 vectors compared to the parental HAdV-C5 vector, though gene expression was only markedly reduced in HAdV-C5/A12. In their in vivo studies, HAdV-C5/A12 proved similarly ineffective in expressing the encoded transgene, and while they concluded that the HAdV-C5/C6 vector was able to evade neutralising antibodies, it was still hampered by HAdV-C5 cross-reactive T cells [264]. 

In 2006, Roberts et al. published their HAdV-C5-based vector, which had all hexon HVRs exchanged for those of HAdV-D48. While it had an impressive ability to evade anti-HAdV-C5 neutralising antibodies in vitro [265], there were significant unpredicted dose-limiting toxicities in subsequent in vivo experiments [266]. Another attempt was made by Tian et al., where the hexon protein of HAdV-B3 was swapped for that of HAdV-B7. Transduction efficiency was not impacted by the hexon swap, and the vector was able to evade neutralising antibodies targeting the parental vector in vitro. In vivo however, there was cross-reactivity to the HAdV-B3/B7 vector in mice immunised with HAdV-B3, possibly due to humoral immunity as well as anti-fibre antibodies [267]. 

In 2018, Nguyen et al. replaced the HVR regions of HAdV-C6 with those of HAdV-C57 in an attempt to create an improved oncolytic platform. While they demonstrated an effective anti-tumour response and designed the vector with reduced anti-vector responses in mind, they did not confirm if their approach had any effect on vector immune evasion [268]. The same vector was also evaluated for use in vaccines in 2019 [269], as well as oncolytic applications in 2022 [270], but without evaluation of the effect of HVR swap on anti-vector immunity. 

The hexon protein selected in both of the latter cases came from the same species, which potentially made it structurally more permissive compared to swaps involving more distantly related Ads, previously attempted. However, it also means amino acid sequence conversion in some of the HVR sites (as well as the rest of the hexon protein), which may act as T cell epitopes. All these studies also highlight that in vitro experiments are not sufficient to accurately predict the effects of hexon modifications in vivo. 

Most recently, Shin et al. published data on their retargeted HAdV-C5-based vector with most or all hexon HVR regions of HAdV-D43. The HVR swapped vector was able to evade anti-HAdV-C5 neutralising antibodies both in vitro and in vivo and showed no cross-reactivity between neutralising antibodies generated [271]. Unfortunately, the administration of the new vector induced a strong anti-vector response, likely making repeated administrations impossible. Furthermore, due to the use of transplanted neutralising serum rather than prime-boost experiments, we currently have no information on whether T cell responses between the parental and new vectors might be cross-reactive [271].

In theory, the change of hexon epitopes, though difficult, presents an attractive way of increasing vector immune evasion against both natural pre-existing immunity (where anti-hexon antibodies have strong neutralising ability, though they are less common) and previous intramuscular vector exposure (where hexon is the immunodominant epitope). To date, attempts have proven unsuccessful and less impactful than expected, possibly due to T cell immunity mostly targeting conserved parts of the hexon protein, making vectors much less efficient in vivo. 

Other studies have investigated whether exchanging the fibre protein may impact the evasion of anti-HAdV-C5 immunity. There has been some evidence that this may aid in the evasion of neutralising immunity. 

A HAdV-C5 vector pseudotyped with the fibre protein of HAdV-D45 published by Parker et al. in 2009 evaded pre-existing immunity in a proportion of tested serum samples unexpectedly well [47]. Another study by White et al. in 2013 with an HAdV-C5-based vector with an HAdV-B35 fibre showed similar results [272]. Both of these studies were performed in human neutralising serum, therefore not providing information on cellular immunity.

Other studies testing samples in immunised mouse serum could not observe the same effect, such as the HAdV-C5 vector with an HAdV-B7 fibre by Gall et al. [273] and Schoggins et al., who also investigated an HAdV-C5 vector with a HAdV-F41 fibre [274], likely due to previously described differences in responses depending on the mechanism of infection, and the anti-hexon skewed response following intramuscular Ad administration. In 2004, Ophorst et al. investigated an HAdV-C5 vector with an HAdV-B35 fibre for suitability as a vaccine vector in vivo in both mice and non-human primates. The effect of the fibre swap on evading anti-vector immunity was described as minimal [275].

Fibre protein is important in endosomal escape, leading to different routes of trafficking in different Ads [276]. Therefore, fibre swaps can change the intracellular trafficking of vectors and the way an anti-vector immune response is mounted intracellularly through the engagement of different PAMPs. Additionally, fibre swaps may have an impact on vectors evading natural pre-existing immunity due to the common pre-existing anti-fibre response.

Though not a major structural protein, pIX is considered a minor capsid protein that is integrated between hexons on the topical side of the capsid [277]. Seregin et al. showed that inserting the complement inhibitor DAF (decay-accelerating factor) in pIX successfully prevented C3 binding, effectively shielding the virus from complement engagement [278]. Disengagement from complement could prevent the induction of anti-Ad neutralising antibodies, though the reduction in immunogenicity would likely come with a major trade-off for vaccine and oncolytic applications.

### 6.2. Chemical Shielding

An advantage of genetic structural alterations, particularly for oncolytic vector use, is the heritability of the modification. Daughter virions produced through replication will all have the same advantages as the parental vector. However, for vectors that do not need to or cannot replicate their immune evasive features, other options are available. A significant body of research into this area concerns the chemical binding and shielding of Ads using hydrophilic polymers. 

Traditionally, polyethene glycol (PEG) derivatives were considered a good option for chemical shielding of Ads. PEG is a long-chain polymer, which, when attached to Ads, will protrude from the capsid in a hairlike manner [279]. Successful examples include O’Riordan [280], Wortmann et al. [281], and Zeng et al. [282], all of whom found that they were able to create shielded vectors able to evade neutralising anti-vector responses in vitro and in vivo. In vivo, PEG shielding also successfully prevented complement activation [196]. 

Unfortunately, PEG itself induces a robust antibody response upon intravenous administration [283]. This limits PEG as a coating option for repeated vector uses. Very high levels of anti-PEG response have been found to be induced by the lipid-nanoparticle-based COVID-19 mRNA vaccine by Pfizer BioNTech, Comirnaty [284]. According to ourworldindata.org, to date, this vaccine has been received by 668 million people across Europe [163]. As a result, at least 73% of the European population (Figure 5) will present anti-PEG-neutralising antibodies, and its use is rapidly becoming obsolete. For Ad vectors specifically there is no longer any advantage of using PEG to shield Ads, and we require more modern solutions. 

Another historically popular shielding polymer is N-(2-hydroxypropyl)methacrylamide (HPMA). HPMA is able to form a multivalent attachment at the capsid surface, coating it much more closely than PEG [279]. HPMA coating has been successful in retargeting Ad vectors from the natural receptor CAR and preventing neutralisation due to pre-existing antibodies in vitro and in vivo [285,286]. It also prevents binding to blood components such as complement and FX [286]. Coating can increase systemic circulation of Ad vector particles by preventing liver uptake for intravenous administrations [287]. HPMA polymers are less immunogenic when administered intravenously [288], making them a good candidate for viral vector coating.

Another current strategy is the use of biodegradable alginate capsules, as done, for example, by Salja et al. Encouragingly, coated vectors were able to evade anti-vector immunity even after HAdV-C5 prime and boost vaccination [289]. An overview of alginate spheres for drug delivery can be found in [290]. A similar strategy employed the use of hyaluronic acid polymers to create vesicles for OV delivery [291]. Though not investigated for immune evasion, packaging of OVs into polymers that the body is already familiar with will very likely be successful in protecting therapeutics from NAbs. This has also been shown by Zhang et al., who coated their HAdV-C5-based vector in extracellular vesicles and showed protection from anti-Ad NAbs in vitro [292].

Alternatively, coating the virus in non-biological materials such as minerals has been investigated, making use of the virus capsid charge to attach differently charged compounds. Wang et al. demonstrated that their vaccine vector was unaffected by anti-vector immunity while still being able to efficiently deliver its transgene [293]. 

The Pluckthun lab has developed a specialised shield, which ablates Ad hexon binding blood factors as well as neutralising antibodies. Additionally, they have developed a “cap” that protects the fibre region, which also ablates any native HAdV-C5 binding to CAR. Though it was mostly developed with Ad retargeting in mind, the combination of the cap and shield showed impressive prevention of neutralisation in vitro, though no in vivo data is currently available [294]. 

An alternative approach combines genetic modifications and chemical shielding. One study demonstrated that the insertion of a cysteine amino acid in HVR5 of the hexon protein enabled the linkage of polymers via thioether links; examples include HPMA (both in a permanent and biodegradable manner) [295] and PEG [296] (which, as described, is no longer very useful beyond proof of concept). However, the potential of this strategy lies in the versatility; as new polymers emerge in time [297], they could likely all be engineered to attach to these Ad vectors. SpyBiotech recently introduced their use of the DogTag/DogCatcher technique for the modification Ads. After genetic insertion of the “tag” into the Ad hexon, they were able to covalently link their “catcher” molecule, which can be bound to any protein or epitope of choice. While it can be used for antigen presentation, it also successfully protects the hexon from FX binding [298]. It is likely that this shielding effect extends to NAbs, though this needs further testing. Despite this interaction being irreversible, the transduction of a GFP transgene in their vector was barely affected [298]. Furthermore, other versions of the tag/catcher system are pH sensitive, so they could be engineered to be biodegradable and shed after vector uptake in the endosome [299]. This system could provide another easy and attractive way of modification of Ad vectors, as the catcher molecule can be bound to a near-infinite number of antigens. 

### 6.3. Other Approaches

A number of alternative approaches have been investigated to address the problem of reducing anti-vector immunity that do not fit naturally into either of the previous categories, some of which have been summarised here. 

Different routes of administration have been proposed as being more or less conducive to Ad transduction in the presence of neutralising antibodies. In 2000, Chen et al. suggested that intramuscular administration of Ad vectors may be able to circumvent anti-Ad immunity when used as a vaccine vector, compared to intranasal and intravenous administration [300]. Controversially, Croyle et al. found that, in the presence of pre-existing anti-vector immunity, only vaccination via the intranasal route, compared to oral and intramuscular, protected mice from lethal dose vaccine-target challenge [301]. Both cases present in vivo studies performed in mice, with initial vector challenge occurring intramuscularly. In 2014, Wu et al. tested their HAdV-C5-based vaccine vector against anti-HAdV-C5 immunity induced intranasally in an effort to emulate natural infection. While they found that their vector had mildly reduced transduction, it still provided protection from vaccine challenge in vivo [302]. Intratumoural injection of Ad-based vectors allows for localised therapeutic effects and activation of both anti-tumour and beneficial anti-Ad immune responses [303]. The unique immunological tumour microenvironment, which is usually immunosuppressed due to the expression of immunomodulatory molecules on cancer cells, may aid in this. However, dissemination of the vector from the site of injection is prevented by preexisting anti-vector immunity [303]. Intradermal administration of an Ad-based vector showed persistent local transgene expression and minimal tissue Ad systemic immune engagement [304]. However, very localised transgene expression such as this will likely have very few clinical uses. A Phase 1 clinical trial of heterologous vaccine schedules with ChAd3 theorised that a longer interval between vaccinations might increase the anti-transgene response, as anti-vector immunity generated in the prime vaccine wanes over time. Pre-existing natural anti-ChAd3 immunity did not affect transgene response after ChAd3 prime vaccination but did increase the anti-vector NAbs generated. This, in turn, reduced anti-transgene responses generated in the boost vaccination. Boosting was more effective if administered after a long interval [305]. In the study, higher anti-vector NAbs were observed pre-prime in the short-interval group compared to the long-interval group. Post-prime, this led to higher concentrations of anti-vector NAbs in the short-interval group, as more anti-vector immunity was present to begin with, making comparison between the groups impossible. Furthermore, the same transgene was used, so a lower amount of antigen was needed for the boost response to be successful. As a result of these experimental limitations, it is not possible to predict how quickly anti-Ad vector immunity recedes over time. However, if it can be confirmed that anti-vector immunity wanes over time within a reasonable timeframe, vectors could be re-used for different transgenes in the future. Case-by-case testing for anti-vector NAb concentrations, though time consuming and potentially expensive, could predict the success of these attempts.

Another interesting approach is the packaging of oncolytic Ad vectors into patient cells is a “Trojan Horse” approach to increase delivery to the target tissue. Iscaro et al. coated oncolytic Ads in liposomes specifically for increased macrophage uptake. For their HAdV-C5/C2-based vector, used in intravenous administration, this strategy was designed to circumvent anti-Ad vector NAbs and destruction by phagocytic cells. Both the virus and a plasmid encoding E1A/B are encapsulated in the liposome to allow for replication in macrophages post-uptake [306]. Macrophages traffic to hypoxic areas of the TME, which are usually hard to reach for intravenous therapies due to the lack of proper blood supply. Through conditional replication of an oncolytic virus under a hypoxia or cancer-specific promoter, OV replication is limited to hard-to-reach areas of solid tumours while also evading neutralizing immunity. Tumour control was also shown in vivo in mice, using a HAdV-C5-based vector in prostate cancer, and NAb evasion was established in vitro in spheroid tumour models using human neutralising anti-Ad serum [306,307]. This approach not only offers the potential for immune evasion of OVs after intravenous application but also additional advantages in the delivery to the tumour site, providing an attractive novel strategy for OV intravenous delivery.

Another strategy involves engineering the vector to infect cells despite the presence of anti-vector immunity by enhancing transduction abilities. In 2023, Yang et al. found that through aggregating their vaccine vector particles and giving them a positive charge through coupling with human serum albumin nanoparticles, transduction in mice was increased both with and without pre-existing vector immunity, compared to the parental vector [308]. Human albumin has been shown previously to protect Ads from NAbs when linked to the hexon protein to coat single virions in vivo in intraperitoneally pre-immunised and intravenously boosted mice [309]. This HAdV-C5-based vector has a mutation in the hexon that allows for independent binding of human serum albumin once in the blood, giving daughter virions the same advantage of biological shielding as parental vectors [310]. This vector is now being investigated in Phase I clinical trials [311,312]. In the case of Yang et al., it is likely that coating with human serum albumin already protected the Ads. However, the aggregation observed by Yang et al. also gave the vector increased transduction ability and helped it transduce despite vaccine-induced vector immunity, rather than evading pre-existing NAbs. 

Generally, replication-deficient or empty capsid vectors would be expected to be less immunogenic than replication-competent vectors in vivo due to the reduced amount of viral protein exposed to the body if no virus progeny is produced [313]. Koup et al. proposed that their replication-deficient HAdV-C5 vector, which contains deletions in several of the early expression genes, failed to induce a CD4+ T cell response, though anti-vector antibodies were detected [314]. Similarly, the deletion of the E2 gene along with the E1 gene of an HAdV-C5-based cancer vaccine increased transgene expression compared to a vector that only had the E1 deletion in the presence of anti-vector immunity [315]. The deletion of the Ad polymerase (E2) reduces late Ad gene expression, therefore reducing the amount of Ad antigen presented on the surface of infected cells.

However, it has also been shown that even after a single dose of replication-deficient HAdV-C5-based vaccine administration, neutralising immunity can be detrimental to repeat administrations [316]. This shows that while deletion of genes for Ad replication likely has some improving effect on generating anti-vector immunity, it will not be enough on its own. 

## 7. Concluding Remarks

Significant progress has been made in the development of more effective Ad-based therapies. Genetic modifications and serotype alternatives to HAdV-C5 have shown promise to improve the evasion of prevalent pre-existing anti-HAdV-C5 immunity. Chemical shielding provides an attractive opportunity to systemically reach sites of interest in the presence of anti-Ad NAbs. Careful selection of vaccination routes and schedules help optimise immunisation strategies for Ad vectors already and close to being in the clinic.

However, Ad-based vector applications have yet to live up to their clinical potential. A part of this may be due to the fact that, as of yet, anti-vector responses are not fully defined or understood, and research in this area has lagged behind the clinical translation of Ad-based therapeutics. As different strategies and routes of administration become more mainstream, our mechanistic knowledge of differences in these routes of administration needs to increase as well. T cell responses to vectors are commonly understudied. While NAb responses are important, ignoring the possibility of cellular anti-vector responses means we do not get the complete picture of what happens in the body upon vector administration. Without this, we will keep seeing misestimations and vectors that work well in an in vitro setting and fail in in vivo systems.

To make better-informed choices of modifications to Ad vectors, we need to better define and understand anti-Ad vector responses overall and to different routes of administration. This understanding can then be applied to the next generation of “smart” Ad-based vector platforms that appropriately evade or harness the immune system where required. This will maximise patient efficacy in the clinic, thus enabling us to use the double-edged sword that is the immune system to benefit rather than limit the patient in the context of Ad-based therapeutics.

## Figures and Tables

**Figure 1 viruses-16-00973-f001:**
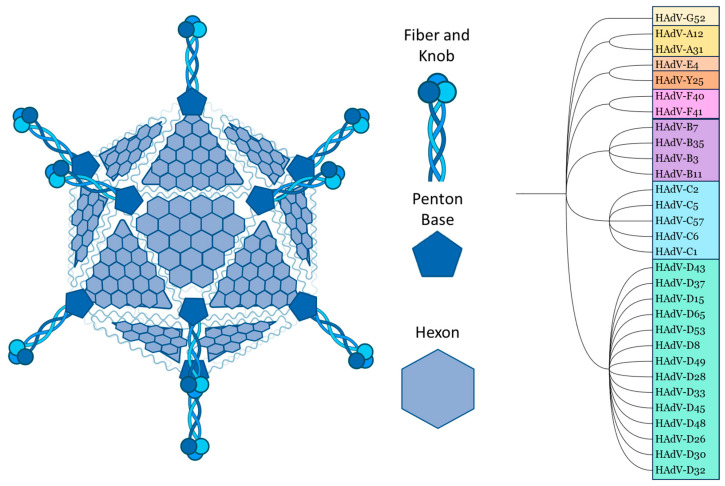
Structural schematic and species classification of selected adenovirus serotypes. Species: A = light orange, B = purple, C = blue, D = green, E/ChAd-Y25 = light/dark red, F = pink, G = yellow. Schematic created using Biorender, PhyloT, and ITOL. All human Ads relevant to this review, as well as chimpanzee (Ch) Ad-Y25, are included in the selection.

**Figure 2 viruses-16-00973-f002:**
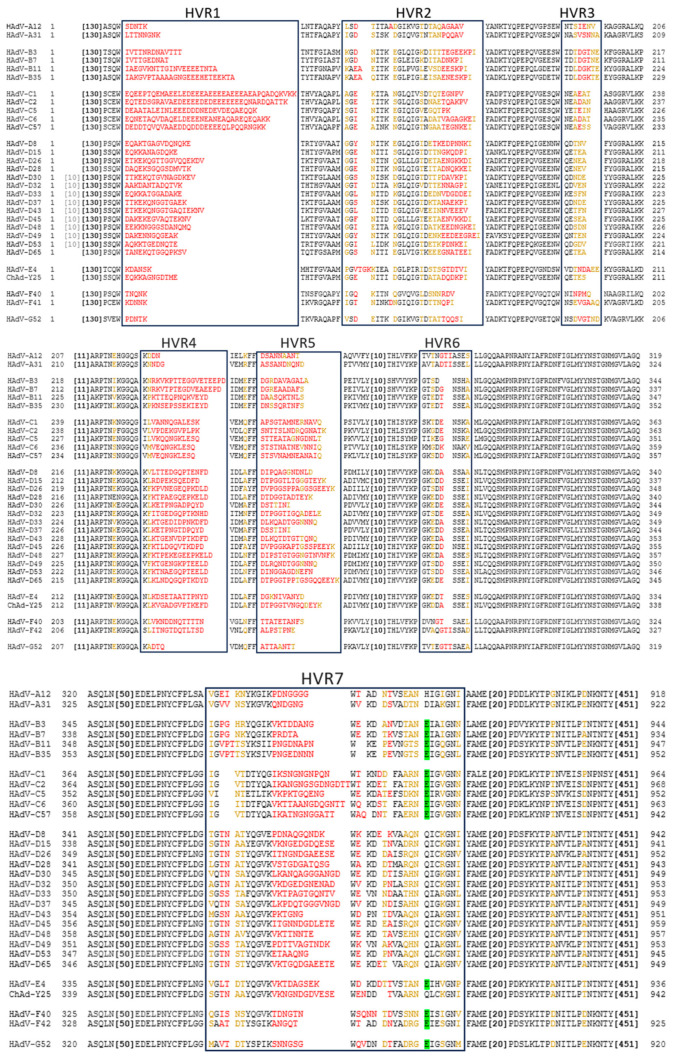
Comparison of Hexon HVRs and FX Binding Sites on Selected Ads of Different Species. Alignment created using NCBI COBALT multiple protein sequence alignment, using NCBI protein IDs. Amino acids in black are highly conserved, in yellow are partially conserved, and in red are hypervariable. The amino acid highlighted in green is most important in FX binding [39]. This model predicts no FX binding across species A and D. Large stretches of conserved sequence have been abbreviated (in brackets). All human Ads relevant to this review, as well as chimpanzee (Ch) Ad-Y25, are included in the selection.

**Figure 3 viruses-16-00973-f003:**
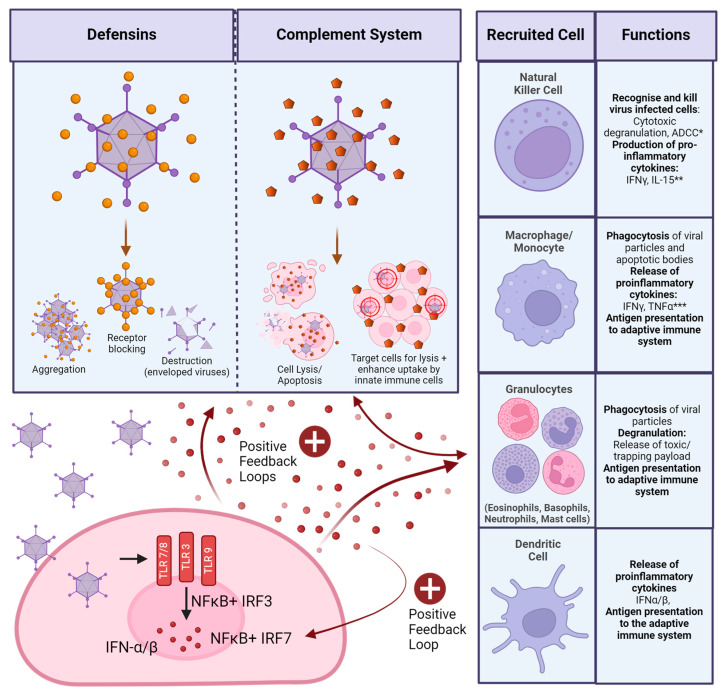
Summary of innate antiviral immune responses. Through sensing via Toll-Like Receptors (TLRs), all cells can secrete IFNs [49]. TLR7/8 senses viral single-stranded RNA, TLR3 senses intracellular viral double-stranded RNA, and TLR9 senses viral double-stranded DNA. Through activation of nuclear factor (NF) κB and IFN regulating factor (IRF) 3 signalling, the cell induces production of IFNs by infected cells. This leads to a positive feedback loop through the auto-activation of IRF7 and the production of more IFN by the cells, leading to a pro-inflammatory response in the infected tissue [48,50]. The secretion of type I IFNs leads to the recruitment of innate immune cells, which have a range of different functions, including the destruction of infected cells and the processing of viral antigens for presentation to the adaptive immune system [50]. The secretion of cytokines serves to further enhance their function and response. Type I IFNs induce upregulation of MHC-I in tissues, as well as APC maturation and increased NK and monocyte/macrophage activation [49]. NK cells recognise the downregulation of MHC-I, which is common in virally infected cells. IFN-γ and IL-15 enhance CD8+ memory responses and upregulate NK cells in a positive feedback loop [49,50]. TNFα induces apoptosis in cells [49]. Blood components also engage the viral pathogen in a multitude of ways, often to enhance the response of innate immune cells. Engagement of complement or defensins may also increase the secretion of IFNs [50,51]. * ADCC: Antibody-dependent cellular cytotoxicity. ** IL: Interleukin. *** TNF: Tumour necrosis factor.

**Figure 4 viruses-16-00973-f004:**
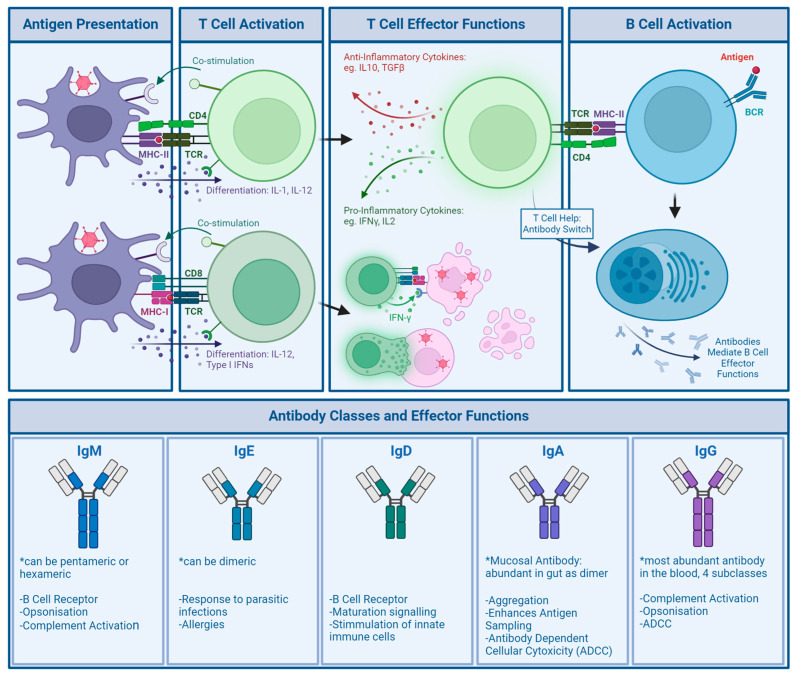
Summary of adaptive antiviral immune responses. Cell activation requires antigen presentation: For CD8+ T cells, this occurs via Major Histocompatibility Complex (MHC) I, and for CD4+ T cells, via MHC-II of professional APCs such as DCs. Generally, 3 signals are required for T cell activation and expansion: Interaction of the TCR with a matching MHC and cognate antigen, co-stimulation (for example, through CD28-CD80/86 in CD4+ T cells), and cytokine signalling [55,56]. Activation induces different effector functions: CD8+ T cells, also called cytotoxic T cells, specialise in either directly killing or inducing apoptosis in infected cells. CD4+ T cells, also called helper T cells, modulate the immune response through the secretion of cytokines. They are also able to activate B cells and induce antibody class switching via CD40/CD40L signalling. Activated B cells turn into plasma cells, which are able to secrete different classes of antibodies, a major component of the adaptive immune response [57,58,59,60,61,62,63]. If B cells are not initially specialised through secretion of antibody types, through class switching, the backbone of secreted antibodies can be changed, changing effector function. After secretion of IgM or IgD as the native B cell receptors, class switching can help specialise the immune response through the secretion of IgG, IgA, or IgE instead [63]. * traits specific to antibodies of this class.

**Figure 5 viruses-16-00973-f005:**
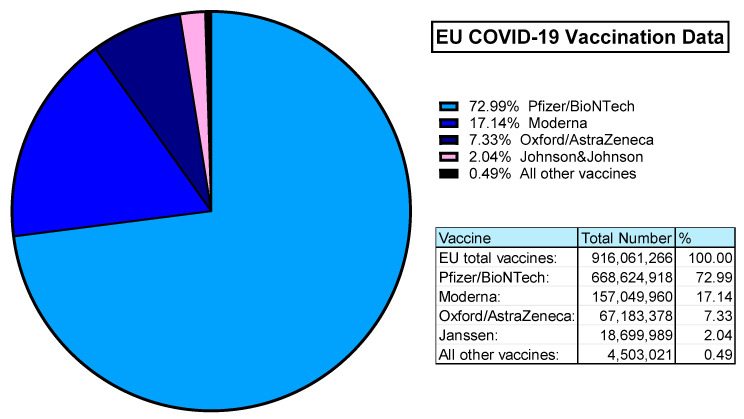
EU COVID-19 Vaccination Data (rounded to 2 decimal places). Data obtained from ourworldindata.org, last accessed 25 March 2024 [163].

**Figure 6 viruses-16-00973-f006:**
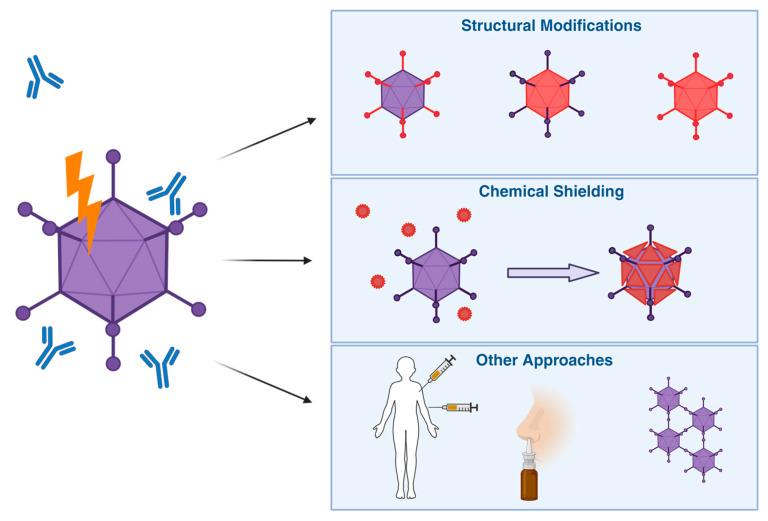
Overview of Different Strategies for Vector Immune Evasion. To help Ad-based vectors evade pre-existing anti-Ad immunity, different solutions have been proposed. These include wholly or partially switching vectors for more rare serotypes, chemical shielding to physically prevent binding of mediators of an immune response, as well as other approaches, such as changes in the route of entry or administration schedules to maximise efficacy despite anti-vector immunity.

**Table 1 viruses-16-00973-t001:** Examples of approved oncolytic and replication-deficient (rep def) E1 deleted Ad-based cancer therapies, as well as current and future clinical trials. Information on clinical trials obtained from https://clinicaltrials.gov/ (accessed 13 June 2024).

Study Title/Drug Name	Status	Mechanism of Action	Vector Based on	Targeted Cancer
Gendicine [134]	CFDA * approved (2003)	Oncolytic, Suicide Gene Therapy (p53 restorative gene therapy)	HAdV-C5	Head and Neck Squamous Cell Carcinoma
Adstiladrin [135]	FDA ** approved (2022)	Immunomodulatory: IFN-α2b Transgene expression	HAdV-C5 (rep def)	Bladder Cancer Unresponsive to First-Line Treatment
First in Human Study With NG-641, a Tumour Selective Transgene Expressing Adenoviral Vector (STAR) [136]	Phase I, recruiting	Oncolytic, Immunostimulatory: FAP-directed bi-specific T-cell activator, Ifα2, CXC9 and CXC10	HAdV-B11p HAdV-B3 (EnAd) [137]	Metastatic or Advanced Epithelial Tumours
First in Man Clinical Study to Evaluate Safety and Tolerability of an Oncolytic Adenovirus in Prostate Cancer Patients [138].	Phase I/IIa, active	Oncolytic	HAdV-C5 [139]	Prostate Cancer
Safety and Efficacy of Repeat Administration of Ad/PNP and Fludarabine Phosphate in Patients With Local Head/Neck Cancer [140]	Phase I/II,	Prodrug Conversion	HAdV-C5 [141] (rep def)	Recurrent Local Head and Neck Cancer
SBRT and Oncolytic Virus Therapy Before Pembrolizumab for Metastatic TNBC and NSCLC (STOMP) [142]	Phase II, active	Prodrug Conversion	HAdV-C5 [143]	Metastatic Triple Negative Breast Cancer, Metastatic Non-Small Cell Lung Cancer
Adenovirus Mediated Suicide Gene Therapy With Radiotherapy in Progressive Astrocytoma [144]	Phase I, recruiting	Prodrug Conversion	HAdV-C5 (rep def)	Different Brain Cancers
Trial Investigating an Immunostimulatory Oncolytic Adenovirus for Cancer [145]	Phase I/II, active	Oncolytic, Immunostimulatory	HAdV-C5/HAdV-B35	Pancreatic Adenocarcinoma, Ovarian Cancer, Biliary Carcinoma, Colorectal Cancer

* CFDA: State Food and Drug Administration of China; ** FDA: U. S. Food and Drug Administration.

**Table 2 viruses-16-00973-t002:** Examples of some current and concluded gene therapy trials using Ad-based vectors. Information obtained from https://clinicaltrials.gov/ (accessed 13 June 2024).

Study Title	Status	Mechanism of Action	Vector Based on	Targeted Disease
Clinical Study With Lymfactin^®^ in the Treatment of Patients With Secondary Lymphedema (AdeLE) [153]	Phase II, not yet recruiting	Transgene: Vascular Endothelial Growth Factor C (VEGF-C)	HAdV-C5 (Lymfactin) [154] (rep def)	Secondary Lymphedema (after radiotherapy for breast cancer)
Epicardial Delivery of XC001 Gene Therapy for Refractory Angina Coronary Treatment (The EXACT Trial) (EXACT) [155]	Phase I/II, completed (2023)	Transgene: VEGF	HAdV-C5 [156,157] (rep def)	Angina caused by coronary artery disease
Preliminary Testing of New Treatment for Chronic Leg Wounds [158]	Phase I, completed (2011)	Transgene: Platelet-Derived Growth Factor B (PDGF-B)	HAdV-C5 (rep def (E1 + E3 deleted [159]))	Chronic wounds of the leg/Venous Ulcers
Phase I Pilot Study of Ad5-CB-CFTR, an Adenovirus Vector Containing the Cystic Fibrosis Transmembrane Conductance Regulator Gene, in Patients With Cystic Fibrosis [160]	Phase I, completed (2001)	Transgene: Cystic Fibrosis Transmembrane conductance Regulator (CFTR)	HAdV-C5 (rep def)	Cystic Fibrosis

**Table 3 viruses-16-00973-t003:** Examples of some vaccines in the clinic as well as current and coming vaccine trials using Ad-based vectors. Information obtained from https://clinicaltrials.gov/ (accessed 13 June 2024).

Study Title/Drug Name	Status	Route of Administration	Vector Based on	Disease
Jcovden	Approved (EMA * March 2021 [170]) Approved (MHRA ** December 2022 [171]) Revoked (FDA *** March 2023 [172])	Intramuscular	HAdV-D26	SARS-CoV-2
Covishield (UK), Vaxzevria(EU)	Approved (EMA * January 2021 [173]) Approved (MHRA ** July 2021 [171])	Intramuscular	ChAdOx1 (ChAd-Y25)	SARS-CoV-2
Phase III Study of BBV154 Intranasal Vaccine in Healthy Volunteers (Nasal154PH3) [174]	Phase III, active	Intranasal	ChAd36 [175]	SARS-CoV-2
Monovalent Chimpanzee Adenoviral-Vectored Marburg Virus Vaccine in Healthy Adults [176]	Phase II, not yet recruiting	Intramuscular	ChAd-3	Marburg Virus
Phase I Clinical Trial With New SARS-CoV-2 CoVacHGMix Type 5 Adenoviral Vector Vaccine [177]	Phase I, recruiting	Intramuscular	HAdV-C5	SARS-CoV-2
Phase 1 Trial of ChAd68 and Ad5 Adenovirus COVID-19 Vaccines Delivered by Aerosol [178]	Phase I, recruiting	Intranasal	ChAd-68 HAdV-C5	SARS-CoV-2
A Clinical Trial on Booster Immunization of Two COVID-19 Vaccines Constructed From Different Technical Routes [179]	Phase I, recruiting	Intranasal	HAdV-C5	SARS-CoV-2
Safety and Immunogenicity of Ad4-HIV Envelope Vaccine Vectors in Healthy Volunteers [180]	Phase I, recruiting	Intranasal, Intramuscular (boost)	HAdV-E4	HIV
Monovalent Chimpanzee Adenoviral-Vectored Marburg Virus Vaccine in Healthy Adults [176]	Phase II, not yet recruiting	Intramuscular	ChAd-3	Marburg Virus

* EMA: European Medicines Agency ** MHRA: Medicines and Healthcare Products Regulatory Agency *** FDA: U. S. Food and Drug Administration.

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
