# Peer review of "The Immune System—A Double-Edged Sword for Adenovirus-Based Therapies"

_viruses, 2024, doi:10.3390/v16060973_

Round 1
Reviewer 1 Report
Comments and Suggestions for Authors
This is an excellent review written by experts in this field. Topics have been well selected and covered in a high level of detail. I have only a few very minor comments to add:
1. In Lines ~725-735 it is argued that homologous boosting with Ad vaccine vectors is likely to be successful despite anti-vector NAbs. I would argue that anti-Ad NAbs do affect homologous boosting of transgene-specific immune responses and are likely a major reason why improvements in immunogenicity upon homologous Ad boosting are lower compared to other technologies such as mRNA and protein vaccines. However, I do accept that this phenomenon is challenging to study mechanistically, particularly in humans. There is some literature around increasing prime-boost intervals to improve boostability (perhaps mitigating the problem of anti-Ad NAbs) (see Capone et al npj vaccines 2020) - perhaps this could be added to the discussion?
2. In the discussion around strategies to shield Ad capsids from NAbs or capsid interactors the authors could mention covalent capsid decoration using Tag/Catcher molecular superglues (see Dicks et al 2022). This approach has been shown to shield Ad capsids in vitro and avoids the challenges of using chemical modification via incorporation of cysteine residues in HVRs (as in cited ref 289) - the latter approach requires preparation of vectors under reducing conditions and storage in an argon atmosphere to control thiol reactivity and prevent virion crosslinking.
3. Line 888 - in my version there appears to be a formatting error here
Author Response
This is an excellent review written by experts in this field. Topics have been well selected and covered in a high level of detail. I have only a few very minor comments to add:
Author response: We thank the reviewer for their positive assessment of our article and for their praise of the detailed content. We appreciate the time taken by the reviewer to review our article and provide their critique.
- In Lines ~725-735 it is argued that homologous boosting with Ad vaccine vectors is likely to be successful despite anti-vector NAbs. I would argue that anti-Ad NAbs do affect homologous boosting of transgene-specific immune responses and are likely a major reason why improvements in immunogenicity upon homologous Ad boosting are lower compared to other technologies such as mRNA and protein vaccines. However, I do accept that this phenomenon is challenging to study mechanistically, particularly in humans. There is some literature around increasing prime-boost intervals to improve boostability (perhaps mitigating the problem of anti-Ad NAbs) (see Capone et al npj vaccines 2020) - perhaps this could be added to the discussion?
Author response: We agree and have added discussion of this paper in 6. Engineering Ads for Immune Evasion à 6.3: Other approaches (lines 1180 – 1196).
- In the discussion around strategies to shield Ad capsids from NAbs or capsid interactors the authors could mention covalent capsid decoration using Tag/Catcher molecular superglues (see Dicks et al 2022). This approach has been shown to shield Ad capsids in vitro and avoids the challenges of using chemical modification via incorporation of cysteine residues in HVRs (as in cited ref 289) - the latter approach requires preparation of vectors under reducing conditions and storage in an argon atmosphere to control thiol reactivity and prevent virion crosslinking.
Author response: We agree and have added discussion of this approach in 6. Engineering Ads for Immune Evasion à 6.2: Chemical Shielding (lines 1146 – 1156).
- Line 888 - in my version there appears to be a formatting error here
Author response: We have corrected this formatting error and thank the reviewer for noting it.
Additional author point: We wish to make the reviewer aware that after the submission of this article we became aware that an article with a similar title already exists (see https://www.frontiersin.org/journals/oncology/articles/10.3389/fonc.2017.00106/full) Although it was not requested, we felt it was prudent to alter the title so it was not so close to this existing manuscript. We hope this is acceptable to the reviewer.
Reviewer 2 Report
Comments and Suggestions for Authors
The review by Wallace and colleagues provides an extensive overview of the complexities associated with the design and clinical use of adenoviral vectors.
This is a really good review that should be a substantial benefit to the adenoviral community at large. I have some suggestions/clarifications (below) to improve the review.
Major points:
1. The basic principles of antiviral immunity in general (as pointed out by the authors on lines 221-223) are well understood. In this regard, the authors may wish to improve the focus of their review by deleting Figures 3 and 4.
2. Lines 71-73. Are the references cited here the most appropriate for the role integrin/RGD in AdV internalization?
3. Figure 1 (legend). Is the NCBI TaxIDs based on DNA sequence or amino acid similarities. Please clarify.
4. Section 3 (lines 163-165) and Figure 3. Another important role for complement is by acting as an opsonin for uptake and removal of viruses by phagocytic cells.
5. Section 4 (lines 228-229). May be worth including the fact that re-emergence of Ad infections in the military resulted from the U.S. government abandoning these vaccines (Kajon, A.E. et al J. Infect. Dis. 2007).
6. Line 300. A good (better) reference for AdV-eye infection/inflammation is Jones, R.A. et al. 2020. Prog. Retin. Eye Res.
7. Section 5.2 (line 614). An appropriate and very recent reference for Ad vectors directly interacting with C3 is Wagner, N. et al. Viruses 15:1343, 2023.
8. Lines 712-714. The text here is somewhat confusing. Did the subjects in this trial have pre-existing ChAd63 Nabs? Wouldn’t this be unexpected?
Minor points:
Line 762 has a typo. “ross-reactivity”
Line 851 typo. “fond”
Line 877 “to be driven by autoantibodies”
Lines 888-889. Heading seems to be out of place.
Line 952. “as they are rare…”
Line 956-957. “ found that a single intranasal vaccination dose conferred protection…”
Line 965. “ The resulting vector…”
Line 973. “They did not look…”
Line 997. Please add reference for Youil et al.
Line 1104. “Another current strategy…”
Line 1130. “ have been investigated to address the problem"
Comments on the Quality of English LanguageSome clarifications are needed as indicated above.
Author Response
The review by Wallace and colleagues provides an extensive overview of the complexities associated with the design and clinical use of adenoviral vectors.
This is a really good review that should be a substantial benefit to the adenoviral community at large. I have some suggestions/clarifications (below) to improve the review.
Author response: We thank the reviewer for their positive critique of our article and for describing it as “extensive” and a “really good review that should be of substantial benefit to the adenoviral community”. We thank the reviewer for their suggested edits which have helped to improve the review.
Major points:
- The basic principles of antiviral immunity in general (as pointed out by the authors on lines 221-223) are well understood. In this regard, the authors may wish to improve the focus of their review by deleting Figures 3 and 4.
Author response: We have considered this point and although we agree that removing these figures may help focus the article, we also feel that the figures are of value to those new to the field. On balance, we prefer to retain these figures and hope the reviewer is content with this.
- Lines 71-73. Are the references cited here the most appropriate for the role integrin/RGD in AdV internalization?
Author response: We agree and apologise for this oversight. We have updated the reference list to cite the appropriate primary references rather than newer and broader reviews.
- Figure 1 (legend). Is the NCBI TaxIDs based on DNA sequence or amino acid similarities. Please clarify.
Author response: We have reworded this figure legend and no longer includes NCBI TaxIDs. This figure is a schematic assembly, not a phylogenetic tree. Information from TaxIDs (species and family information) was used but is not required to generate this figure. We agree that the original wording was confusing and did not make this clear.
- Section 3 (lines 163-165) and Figure 3. Another important role for complement is by acting as an opsonin for uptake and removal of viruses by phagocytic cells.
Author response: We agree, and thank the reviewer for reminding of to include mention of this important role of complement. We have noted this in the revised text (line 167-8) and in figure 3.
- Section 4 (lines 228-229). May be worth including the fact that re-emergence of Ad infections in the military resulted from the U.S. government abandoning these vaccines (Kajon, A.E. et al J. Infect. Dis. 2007).
Author response: We agree and have added a short sentence covering this point in the revised manuscript (line 232-235)
- Line 300. A good (better) reference for AdV-eye infection/inflammation is Jones, R.A. et al. 2020. Retin. Eye Res.
Author response: We agree and have added this reference (ref 91), thank you!
- Section 5.2 (line 614). An appropriate and very recent reference for Ad vectors directly interacting with C3 is Wagner, N. et al. Viruses15:1343, 2023.
Author response: We have included this reference in the revised manuscript (ref 194)
- Lines 712-714. The text here is somewhat confusing. Did the subjects in this trial have pre-existing ChAd63 Nabs? Wouldn’t this be unexpected?
Author response: We have added a few sentences to clarify. Yes, neutralising antibodies to ChAd were observed and are unexpected. We have expanded text in lines 722-724 to address this, with some additional comments on lines 749-750, as this is not a unique observation. Cross-reactivity has been suggested as a reason for the observation of high levels of anti-ChAd NAbs.
Minor points:
Line 762 has a typo. “ross-reactivity”
Line 851 typo. “fond”
Line 877 “to be driven by autoantibodies”
Lines 888-889. Heading seems to be out of place.
Line 952. “as they are rare…”
Line 956-957. “ found that a single intranasal vaccination dose conferred protection…”
Line 965. “ The resulting vector…”
Line 973. “They did not look…”
Line 997. Please add reference for Youil et al.
Line 1104. “Another current strategy…
Line 1130. “ have been investigated to address the problem"
Author response: We have corrected all of the above typos. We thank the reviewer for noting them.
Additional author point: We wish to make the reviewer aware that after the submission of this article we became aware that an article with a similar title already exists (see https://www.frontiersin.org/journals/oncology/articles/10.3389/fonc.2017.00106/full) Although it was not requested, we felt it was prudent to alter the title so it was not so close to this existing manuscript. We hope this is acceptable to the reviewer.